# Retrieval of aerosol micro-physical and optical properties over land using a multi-mode approach

Guangliang Fu[1] and Otto Hasekamp[1]

[1]Netherlands Institute for Space Research (SRON, NWO-I), Utrecht, The Netherlands.

**Correspondence:** Guangliang Fu (g.fu@sron.nl)

**Abstract.** Polarimeter retrievals can provide detailed and accurate information on aerosol microphysical and optical properties. The SRON-Aerosol algorithm is one of the few retrieval approaches that can fully exploit this information. The algorithm core is a 2-mode retrieval where effective radius ($r_{\text{eff}}$), effective variance ($v_{\text{eff}}$), refractive index, column number are retrieved for each mode, the fraction of spheres for the coarse mode, and an aerosol layer height. Further, land and ocean properties are retrieved simultaneously with the aerosol properties. In this contribution, we extend the SRON-Aerosol algorithm by implementing a multi-mode approach where each mode has fixed $r_{\text{eff}}$ and $v_{\text{eff}}$. In this way the algorithm gets more flexibility in describing the aerosol size distribution and avoids the high non-linear dependence of the forward model on the aerosol size parameters. On the other hand, the approach depends on the choice of the modes.

We compare the performances of multi-mode retrievals (varying the number of modes from 2 to 10) with those based on the original (parametric) 2-mode approach. Experiments with both synthetic measurements and real measurements (PARA-SOL satellite level-1 data of intensity and polarization) are conducted. The synthetic data experiments show that multi-mode retrievals are good alternatives to the parametric 2-mode approach. It is found that for multi-mode approaches, with 5 modes the retrieval results can be already good for most parameters. The real data experiments (validated with AERONET data) show that, for the Aerosol Optical Thickness (AOT), multi-mode approaches achieve higher accuracy than the parametric 2-mode approach. For Single Scattering Albedo (SSA), both approaches have similar performances.

## 1  Introduction

Aerosols such as dust, smoke, sulphate, and volcanic ash affect the Earth's climate by interaction with radiation (direct effect) and by modifying the properties of clouds (indirect effect). In order to reduce the large uncertainties in aerosol direct and indirect effect, satellite remote sensing is of crucial importance (Lee et al., 2009). Satellite data of intensity and polarization (polarized intensity) that observe a ground pixel under multiple viewing angles contain the richest set of information of aerosols in our atmosphere from a passive remote sensing perspective (Kokhanovsky, 2015). To acquire useful knowledge based on these data, accurate retrievals of aerosols' microphysical and optical properties are essential. Here, aerosol microphysical properties include the particle effective radius, the effective variance, the refractive index and the particle shape. Aerosol optical properties mainly include the (multi-spectral) aerosol optical thickness and single scattering albedo. Accuracy requirements for (a subset of) these parameters are listed in Table 1.

There are currently a number of aerosol retrieval algorithms available (Chowdhary et al., 2001; Hasekamp and Landgraf, 2007; Hasekamp, 2010) based on the use of multi-angle and multi-spectral measurements of intensity and polarization. These algorithms can be divided in two main groups: LookUp-Table (LUT) based approaches and full inversion approaches. Generally speaking, LUT approaches are faster but less accurate than full inversion approaches because LUT approaches choose the best fitting aerosol model from a discrete lookup table. Full inversion approaches are more accurate but slower because they require radiative transfer calculations as part of the retrieval procedure. The LUT algorithms are e.g., the LOA LUT algorithm over ocean (Deuzé et al., 2000), the LOA LUT algorithm over land (Deuzé et al., 2001; Herman et al., 1997), and the SSA LUT algorithm (Waquet et al., 2016). The full inversion algorithms are e.g., the GRASP algorithm (Dubovik et al., 2011), the SRON-Aerosol algorithm (Hasekamp and Landgraf, 2007; Hasekamp et al., 2011; Stap et al., 2015; Wu et al., 2015, 2016; Di Noia et al., 2017), the JPL algorithm (Xu et al., 2017), the GISS algorithm (Waquet et al., 2009) and the MAPP algorithm (Stamnes et al., 2018). Besides, some additional aerosol retrieval approaches can be found in (Sano et al., 2006; Cheng et al., 2011; Masuda et al., 2000; Lebsock et al., 2007). It should be noted that of the full inversion approaches only the SRON-Aerosol algorithm and the GRASP algorithm have been applied at a global scale.

In this study, the SRON-Aerosol algorithm is used, which is a full inversion retrieval approach with the first guess generated by LUT retrieval. In the SRON-Aerosol algorithm, a damped Gauss-Newton iteration method is used to solve the non-linear retrieval problem. Phillips-Tikhonov regularization is used as the regularization method. In the current version of the algorithm, it is based on a bi-modal description of aerosols in a fine- and coarse mode respectively, both described by a log-normal size distribution. The parameters that describe these 2 modes (for each mode $r_{\text{eff}}$, $v_{\text{eff}}$, refractive index, column number and for the coarse mode additionally the fraction of spheres) are being retrieved. A similar approach has been used by Waquet et al. (2009) and Stamnes et al. (2018). Other algorithms (GRASP, JPL) do not retrieve size parameters of each mode but instead describe aerosols with a larger number of modes with fixed size distribution. The column number of each mode is then a free parameter in the retrieval.

Both approaches have advantages and disadvantages. The bi-modal approach may not be appropriate in situations where aerosols contain more than 2 modes. Also, the retrieval of $r_{\text{eff}}$ and $v_{\text{eff}}$, of each mode makes the inversion problem highly non-linear and hence more difficult to solve. On the other hand, multi-mode approaches are expected to depend strongly on the assumed size distribution of each mode and the total number of modes used.

The aim of this paper is to compare the bi-modal and multi-modal approaches for the retrieval of aerosols from Multi-Angle Polarimeter (MAP) data. For this purpose we extend the SRON algorithm with the capability to do a multi-mode retrieval. We then compare the approaches for synthetic measurements and for real measurements of POLDER-3 on PARASOL.

This paper is organized as follows. Sect. 2 introduces the methodologies of the parametric 2-mode retrieval and multi-mode retrievals. Sect. 3 describes the data sets and retrieval quality measures used in this study. Sect. 4 contains the synthetic data experiments. The real data experiments of multi-mode approaches are discussed in Sect. 5. Finally, the last section summarizes and concludes this study.

## 2 Methodology

### 2.1 Parametric 2-mode retrieval

In this section, we first describe the methodology of the original SRON-Aerosol algorithm, which is is referred to as a parametric 2 mode retrieval. The inversion retrieval approach is aimed to invert a forward model equation:

$$\boldsymbol{y} = \mathbf{F}(\boldsymbol{x}) + \boldsymbol{e}_y. \tag{1}$$

Here, $\boldsymbol{y}$ is the measurement vector containing the multi-spectral and multi-angle polarimetric measurements of PARASOL. $\boldsymbol{e}_y$ represents the measurement error. $\boldsymbol{x}$ contains parameters to be retrieved, which includes aerosol properties and land or ocean properties. The forward model $\mathbf{F}(\boldsymbol{x})$ which describes the dependence between $\boldsymbol{y}$ and $\boldsymbol{x}$ contains two parts: (1) microphysical properties to optical properties; (2) optical properties to the intensity vector (at the top of the atmosphere) through an atmospheric Radiative Transfer model (RT). Nonspherical aerosols are modeled as a size/shape mixture of randomly oriented spheroids (Hill et al., 1984; Mishchenko et al., 1997). We use the Mie/T-Matrix/Improved-Geometrical-Optics database by Dubovik et al. (2006) along with their proposed spheroid aspect ratio distribution for computing optical properties for a mixture of spheroids and spheres. For the RT we refer to Landgraf et al. (2001) and Hasekamp and Landgraf (2002); Hasekamp and Landgraf (2005).

In the parametric 2-mode retrieval algorithm, the fine/coarse mode (denoted by superscript $^{\text{f}}$ or $^{\text{c}}$) is characterized by the effective radius $r_{\text{eff}}^{\text{f;c}}$, the effective variance $v_{\text{eff}}^{\text{f;c}}$, the real and imaginary part of refractive index $m_{\text{r}}^{\text{f;c}}$ and $m_{\text{i}}^{\text{f;c}}$, the aerosol loading $N^{\text{f;c}}$ and the fraction of spheres $f_{\text{sphere}}^{\text{f;c}}$. The complex refractive index for each mode is $m^{\text{f;c}} = m_{\text{r}}^{\text{f;c}} + i m_{\text{i}}^{\text{f;c}}$. In the latest SRON-Aerosol algorithm, $m^{\text{f;c}}$ are not directly retrieved (i.e., not in the state vector $\boldsymbol{x}$), but constructed by $m^{\text{f;c}}(\lambda) = \sum_{k=1}^{n_{\alpha}^{\text{f;c}}} \alpha_k^{\text{f;c}} m^{k,\text{f;c}}(\lambda)$ where the mode component coefficients $\alpha_k^{\text{f;c}}$ ($0 \leq \alpha_k^{\text{f;c}} \leq 1$) are included in the retrieval state vector. $m^{k,\text{f}}$ for the fine mode (or $m^{k,\text{c}}$ for the coarse mode) are the fixed spectral-dependent complex refractive index spectra for some aerosol components, e.g., DUST, water (H2O), Black Carbon (BC), INORGanic matter (INORG). In this study, we set $n_{\alpha}^{\text{f;c}} = 2$ and assume that the fine mode and the coarse mode are respectively composed by INORG+BC and DUST+INORG. Note that this assumption is flexible and can be updated according to the information content of the measurement. Also spectra based on Principal Component Analysis (PCA) can be used like in (Wu et al., 2015).

To retrieve the state vector from the satellite measurements, a damped Gauss-Newton iteration method with Phillips-Tikhonov regularization is employed (Hasekamp et al., 2011). The inversion algorithm finds the solution $\hat{\boldsymbol{x}}$ which solves the minimization/optimization problem,

$$\hat{\boldsymbol{x}} = \min_{\boldsymbol{x}}(||\mathbf{S}_y^{-\frac{1}{2}}(\mathbf{F}(\boldsymbol{x}) - \boldsymbol{y})||^2 + \gamma ||\mathbf{W}^{-\frac{1}{2}}(\boldsymbol{x} - \boldsymbol{x}_a)||^2). \tag{2}$$

Here, $\boldsymbol{x}_a$ is the a priori state vector, $\mathbf{W}$ is a weighting matrix, $\gamma$ is a regularization parameter, and $\mathbf{S}_y$ is the measurement error covariance matrix. The weighting matrix $\mathbf{W}$ ensures that all state vector parameters range within the same order of magnitude (Hasekamp et al., 2011) and can be used to give some parameters more freedom in the inversion than others (similar to the prior covariance matrix in Optimal Estimation Methods). Since the forward model $\mathbf{F}(\boldsymbol{x})$ is nonlinear with respect to $\boldsymbol{x}$, the

inversion for Eq. (2) is implemented iteratively. For each iteration step (e.g., step $n$), we approximate the forward model $\mathbf{F}(\boldsymbol{x})$ with

$$\mathbf{F}(\boldsymbol{x}) \approx \mathbf{F}(\boldsymbol{x}_n) + \mathbf{K}(\boldsymbol{x} - \boldsymbol{x}_n). \tag{3}$$

Here, $\mathbf{K}$ is the Jacobian matrix (with $K_{ij} = \frac{\partial F_i}{\partial x_j}(\boldsymbol{x}_n)$), which contains the derivatives of the forward model with respect to each variable in the state vector $\boldsymbol{x}$.

Based on the linear approximation (Eq. (3)), the optimization problem (Eq. (2)) can be reduced to

$$\tilde{\boldsymbol{x}}_{n+1} = \min_{\tilde{\boldsymbol{x}}}(||\tilde{\mathbf{K}}(\tilde{\boldsymbol{x}} - \tilde{\boldsymbol{x}}_n) - \tilde{\boldsymbol{y}}||^2 + \gamma||\tilde{\boldsymbol{x}} - \tilde{\boldsymbol{x}}_a||^2), \tag{4}$$

where $\tilde{\mathbf{K}} = \mathbf{S}_y^{-\frac{1}{2}}\mathbf{K}\mathbf{W}^{\frac{1}{2}}$, $\tilde{\boldsymbol{x}} = \mathbf{W}^{-\frac{1}{2}}\boldsymbol{x}$ and $\tilde{\boldsymbol{y}} = \mathbf{S}_y^{-\frac{1}{2}}(\boldsymbol{y} - \mathbf{F}(\boldsymbol{x}_n))$. The solution of Eq. (4) refers to Rodgers (2000); Hasekamp et al. (2011) and is iterated by:

$$\tilde{\boldsymbol{x}}_{n+1} = \Lambda\tilde{\mathbf{G}}\tilde{\boldsymbol{y}} + \tilde{\mathbf{A}}\tilde{\boldsymbol{x}}_n + (\mathbf{I} - \tilde{\mathbf{A}})\tilde{\boldsymbol{x}}_a, \tag{5}$$

with the contribution matrix $\tilde{\mathbf{G}} = (\tilde{\mathbf{K}}^T\tilde{\mathbf{K}} + \gamma\mathbf{I})^{-1}\tilde{\mathbf{K}}^T$ and the averaging kernel matrix $\tilde{\mathbf{A}} = \tilde{\mathbf{G}}\tilde{\mathbf{K}}$. $\Lambda$ is a filter/damping factor, which limits the step size for each iteration of the state vector. In this way, we use a Gauss-Newton scheme with reduced step size to avoid diverging retrievals (Hasekamp et al., 2011). The filter factor $\Lambda$ values between 0 and 1.

The regularization parameter $\gamma$ and filter factor $\Lambda$ in Eqs. (4) and (5) are chosen optimally (for each iteration) from different values for $\gamma$ (10 values from 0.1 to 5) and for $\Lambda$ (10 values from 0.1 to 1) by evaluating the goodness-of-fit using a simplified (fast) forward model.

## 2.2 Multi-mode retrieval

We now introduce the multi-mode SRON-Aerosol retrieval approach. In principle, the idea of the multi-mode approach is that instead of fitting the size distribution parameters ($r_{\text{eff}}$ and $v_{\text{eff}}$) of two modes, one aims to fit the size distribution with a larger number of modes for which $r_{\text{eff}}$ and $v_{\text{eff}}$ are fixed. An expected advantage of this approach is that it makes the inversion problem more linear ($r_{\text{eff}}$ and $v_{\text{eff}}$ tend to make the inversion problem highly non-linear). Furthermore, the multi-mode approach has more freedom in fitting different shapes of size distribution if the number of chosen modes is sufficiently large. On the other hand, the multi-mode approach is expected to depend strongly on the assumed modes.

The performance of multi-mode are expected to be better and better as the mode number increases. In this study, we take the 10-mode retrieval as the maximum mode-number retrieval. All the multi-mode retrieval cases are defined as in Table 2. For example, for the 5-mode retrieval case, the 5 modes used for retrieval are actually mode 2, 4, 6, 7, 9 in the 10-mode retrieval. Here, the 5 modes correspond to those of Xu et al. (2017). The abbreviations for different retrieval cases used in this study are listed in Table 3, where the parametric retrieval is denoted with a superscript [c], i.e., 2modeRetr[p].

For multi-mode retrievals, the state vector $\boldsymbol{x}$ in Eqs. (1) and (2) is different from that in the parametric 2-mode retrieval. The difference is shown in Table 4, which specifies the parameters in the state vector. In the multi-mode retrieval, since $r_{\text{eff}}$ and $v_{\text{eff}}$ are not retrieved for all modes, thus they are not included in $\boldsymbol{x}$. The aerosol loading $N^j (j = 1, 2, ..., n_{\text{mode}})$ for all

modes are retrieved and included in $x$. In principle, other aerosol parameters like the refractive index coefficients, the fraction of spheres, and the aerosol layer height can be retrieved for each mode independently. However, the measurement vector will not contain sufficient information to extract this information for each mode separately. Therefore, for the retrieval of these parameters we group the modes into 2 types - fine (i.e., mode 1-6 of of Table 2 for the 10-mode case) and coarse (mode 7-10 of Table 2). For the refractive index coefficients, we fit one value for the fine modes and one value for the coarse modes. For the fraction of spheres, we only retrieve one value for the coarse modes and assume the fine modes consist only of spheres. (A recent study by Liu and Mishchenko (2018) indicates that this assumption becomes unrealistic for increasing fraction of carbonaceous aerosol in the fine mode.) For the aerosol layer height we fit one value that is assumed representative for all modes. These assumptions are similar to those in the parametric 2-mode retrieval. According to Table 4, the number of aerosol parameters for the parametric 2-mode retrieval and the multi-mode retrieval is respectively $n_\alpha^{\mathrm{f}}+n_\alpha^{\mathrm{c}}+8$ and $n_{\mathrm{mode}}+n_\alpha^{\mathrm{f}}+n_\alpha^{\mathrm{c}}+2$, where the fine and coarse mode component coefficients $n_\alpha^{\mathrm{f}}$ and $n_\alpha^{\mathrm{c}}$ are both set to 2 in this study.

Besides the aerosol-related parameters, $x$ in multi-mode retrievals also includes surface reflectance and polarization parameters in the same manner as the parametric 2-mode retrieval. For surface models of the Bidirectional Reflectance Distribution Function (BRDF), we use the ROSS-LI model (Li and Strahler, 1992; Ross, 1981) for the same settings as in Litvinov et al. (2011). For modeling surface Bidirectional Polarization Distribution Function (BPDF), a Fresnel model is used as introduced by Maignan et al. (2009). The surface parameters, to be retrieved in the state vector (see Table 4), are a scaling parameter for BPDF model ($x_{\mathrm{bpdf}}^{\mathrm{scale}}$), the coefficient of the LI sparse kernel ($x_{\mathrm{bdrf}}^{\mathrm{geo1}}$), the coefficient of the ROSS thick kernel ($x_{\mathrm{bdrf}}^{\mathrm{geo2}}$), and the BDRF scaling parameters at each wavelength band ($x_{\mathrm{bdrf}}^{iw}$, $iw = 1, 2, \cdots, n_{\mathrm{wave}}$). The number of surface-related parameter in the state vector for all retrieval cases is $n_{\mathrm{wave}}+3$. Therefore, the length of the state vector (i.e., the total number of aerosol- and surface-related parameters) is $n_\alpha^{\mathrm{f}}+n_\alpha^{\mathrm{c}}+n_{\mathrm{wave}}+11$ for the parametric 2-mode retrieval, and is $n_{\mathrm{mode}}+n_\alpha^{\mathrm{f}}+n_\alpha^{\mathrm{c}}+n_{\mathrm{wave}}+5$ for the multi-mode retrieval.

The inversion procedure of multi-mode retrievals is the same as described by Eqs. (3), (4) and (5). $\mathbf{W}$ is a diagonal matrix and its diagonal values are shown in Table 5. Note that, the prior information of aerosol loading ($N$) is provided in terms of AOT.

## 2.3   Multi-mode retrieval of first guess

In the SRON-Aerosol algorithm, the first guess of $x$ is obtained before the full inversion retrieval using a lookup table, which is based on tabulated radiative transfer calculations for each of the 10 modes listed in Table 6 separately. The radiative transfer calculations are performed for different combinations of input parameters (as specified in Table 6) which are e.g., one single layer height, one value of the refractive index (different for fine and coarse modes), 9 AOT ($\tau$) values, 15 wavelength bands, 7 Viewing Zenith Angles (VZA), 14 Solar Zenith Angles (SZA), 2 surface pressures, 2 values for the scaling parameter for BPDF model, 3 values for the coefficient of LI Sparse kernel, 4 values for the coefficient of ROSS Thick kernel, 7 values for the BDRF scaling parameters at each wavelength band.

The pre-calculated LUT is used as input for an approximate forward model in the LUT retrieval. Here, the radiative transfer multiple scattering calculations, performed separately for the different modes, are combined using the method of Gordon and

Wang (1994). Single scattering is computed exactly as its computational cost is negligible. Using the approximate forward model, a retrieval is performed using the same inversion method as for the full retrieval (Eqs. (3)–(5)). The fit parameters are the aerosol column numbers of the 10 modes and the surface parameters. The result of the 10-mode LUT retrieval is also used for full retrievals with less than 10 modes (e.g. the parametric 2 mode retrieval), by fitting the $n_{\mathrm{mode}}$ ($n_{\mathrm{mode}} < 10$) size distribution to the 10-mode size distribution coming from the LUT retrieval with the $n_{\mathrm{mode}}$ aerosol columns for the different modes as fit parameters.

## 3 Data and retrieval measures

### 3.1 PARASOL data

The satellite data used in this study for aerosol retrievals are from the Polarization and Directionality of Earth Reflectances-3 (POLDER-3) instrument (Deschamps et al., 1994; Fougnie et al., 2007), which was mounted on the PARASOL satellite (retired in 2013). The POLDER-3 instrument in space provided in orbit multiangle and multispectral photopolarimetric measurements of intensity and polarization. The PARASOL level-1 Collection 3 product data have been used in this study.

Each PARASOL image including $242 \times 274$ elements was made on a CCD matrix array over a total view of $114°$. Each ground pixel (6 km $\times$ 6 km) is measured under up to 16 angles. The intensity component (Stokes parameter $I$) was measured at 9 bands and the polarization component (Stokes parameters $Q$ and $U$) was measured at 490, 670 and 865 nm. PARASOL has a swath width of about 2400 km. The data from PARASOL have been used for aerosol retrievals in a number of studies (Dubovik et al., 2011; Hasekamp et al., 2011; Stap et al., 2015; Lacagnina et al., 2015, 2017). In previous studies using the SRON-Aerosol algorithm (Hasekamp et al., 2011; Stap et al., 2015; Lacagnina et al., 2015, 2017), four bands (i.e., 490, 670, 865, 1020 nm) were used. In this study, two more bands (440 and 565 nm) are added for retrievals.

In the SRON-Aerosol algorithm, we do not directly use $Q$ and $U$ in the measurement vector, but use the Degree of Linear Polarization (DoLP) as the polarization component (together with the intensity component $I$) in the measurement vector. Here, DoLP equals to $\frac{\sqrt{Q^2+U^2}}{I}$. For our retrievals on PARASOL measurements, we assume an intensity error $I_{\mathrm{err}} = 0.01$ and the polarization error $\mathrm{DoLP_{err}} = 0.007$, in the diagonal matrix $\mathbf{S}_y^{\frac{1}{2}}$ in Eq. (4). Here the intensity error is the relative error, and the polarization error is the absolute error. $I_{\mathrm{err}} = 0.01$ holds for all POLDER bands except for the band 440 nm, where $I_{\mathrm{err}}^{440}$ is set at 0.03 because the intensity measurements at 440 nm are usually considered less accurate than those at other bands (Fougnie et al., 2007; Dubovik et al., 2011). Note that in our study in principle 0.01 for $I_{\mathrm{err}}$ and 0.007 for $\mathrm{DoLP_{err}}$ used are underestimating the PARASOL errors but in our inversion approach only the relative dependence between intensity errors and Degree of Linear Polarization (DoLP) errors is important. The absolute value is compensated by the regularization parameter.

It should also be noted that higher accuracy aerosol retrievals are to be expected for all parameters from instruments that have higher polarimetric accuracy, more scattering angles and/or spectral bands (e.g. (Mishchenko and Travis, 1997; Hasekamp and Landgraf, 2007)). Examples of such improved instruments are GLORY-APS (Mishchenko et al., 2007), MAIA (Diner et al., 2018), SPEXone (Hasekamp et al., 2018), and HARP-2 (Martins et al., 2017).

## 3.2 Meteorological data

During retrievals, some atmospheric and meteorological inputs are needed to be interpolated to each pixel (where there is a PARASOL measurement) at a specified time and a geographical location. The required atmospheric parameters/inputs are humidity, temperature, pressure, height. In this study, we obtain these information from National Centers for Environmental Prediction (NCEP) reanalysis data (Kalnay et al., 1996).

## 3.3 AERONET data

In this study we focus on aerosol retrievals over land. We validate the retrieved Aerosol Optical Thickness (AOT) with AERONET (AErosol RObotic NETwork) level 2.0 data (quality-assured) of AOT (Holben et al., 2001). The retrieved Single Scattering Albedo (SSA) is validated with AERONET level 1.5 (cloud-screened and quality controlled) Almucantar Retrieval Inversion Products (Dubovik et al., 2002) of SSA.

## 3.4 Retrieval measures

In a retrieval, it is a common approach to apply the goodness of fit ($\chi^2$) to decide whether the retrievals have successfully converged. The goodness of fit $\chi^2$ for each pixel is calculated by:

$$\chi^2 = \frac{1}{n_{\text{meas}}} \sum_{i=1}^{n_{\text{meas}}} \frac{(F_i - y_i)^2}{S_y(i,i)}. \tag{6}$$

Here, $n_{\text{meas}}$ is the total number of measurements (multi-angle and multi-spectral intensity and polarization) for each pixel. $y_i$ represents the measurement (synthetic or real) and $F_i$ represents the simulated measurement through the forward model. $S_y(i,i)$ is the diagonal value of the measurement error covariance matrix, corresponding to the $i^{\text{th}}$ measurement.

We consider retrievals with $\chi^2 < \chi^2_{\text{max}}$ as valid retrievals. This filter rejects cases where the forward model is not able to fit the measurements, i.e. because of cloud contaminated pixels (Stap et al., 2015, 2016), corrupted measurements (Hasekamp et al., 2011), and cases where the first guess state vector deviates too much from the truth. Based on $\chi^2$, we define the pass rate $r_{\text{pass}} = \frac{n_{\text{pass}}}{n_{\text{pix}}}$ to be the number of successful pixels ($n_{\text{pass}}$) over the number of all pixels ($n_{\text{pix}}$).

To evaluate the retrieved aerosol properties, two measures are used, which are the Root Mean Square Error (RMSE) and the bias. The two measures are both with respect to the differences between the retrieved values and the reference values (AERONET for real measurements and the truth for synthetic measurements). Here the difference $x_{\text{ipix}}^{\text{diff}}[j]$ (at the pixel ipix for the $j^{\text{th}}$ variable in the state vector $x$) is computed by $x_{\text{ipix}}^{\text{diff}}[j] = x_{\text{ipix}}^{\text{retr}}[j] - x_{\text{ipix}}^{\text{true}}[j]$, where $x_{\text{ipix}}^{\text{retr}}$ represents the retrieved aerosol property for the pixel ipix, while $x_{\text{ipix}}^{\text{true}}$ represents the reference aerosol property.

For each aerosol property, The RMSE counts the overall retrieval errors for all pixels by $\sqrt{\frac{1}{n_{\text{pass}}} \sum_{\text{ipix}=1}^{n_{\text{pass}}} (x_{\text{ipix}}^{\text{diff}}[j])^2}$. The bias is calcualed by $\frac{1}{n_{\text{pass}}} \sum_{\text{ipix}=1}^{n_{\text{pass}}} (x_{\text{ipix}}^{\text{diff}}[j])$. The bias can be positive or negative, meaning the overestimation or the underestimation.

## 4 Synthetic retrievals

### 4.1 Synthetic measurements

To investigate the capability of multi-mode retrievals of aerosol microphysical and optical properties, we first perform synthetic data experiments. We can assess the capability of different retrieval setups by comparing the result of the retrieval to the "truth" that was used to create the synthetic measurement. The synthetic measurements are computed for the PARASOL wavelengths and 14 viewing angles which is representative for PARASOL (Sect. 3.1).

The synthetic measurements are created pixel by pixel with two steps: (1) Generating aerosol modes based on assumed true aerosol properties of the effective radius $r_{\mathrm{eff}}$, the effective variance $v_{\mathrm{eff}}$, the fraction of spheres $f_{\mathrm{sphere}}$, the aerosol loading $N$, the mode component coefficients $\alpha_k$, the aerosol height $z$. In this study, two sets of synthetic measurements are created. One set is created based on 10 aerosol modes. Each mode has fixed $r_{\mathrm{eff}}$, $v_{\mathrm{eff}}$ as shown in Table 2. The other set is 2-mode based. For this set, $r_{\mathrm{eff}}^{\mathrm{f}}$ and $r_{\mathrm{eff}}^{\mathrm{c}}$ are perturbed within [0.1, 0.3] and [0.65, 3.4], respectively. $v_{\mathrm{eff}}^{\mathrm{f}}$ and $v_{\mathrm{eff}}^{\mathrm{c}}$ are perturbed within [0.1, 0.3] and [0.4, 0.6], respectively. (2) Based on the generated aerosol modes, the forward model as discussed in Sect. 2.1 is used to generate the synthetic measurements. The assumed true aerosol properties for each pixel are generated stochastically.

For synthetic data experiments, we only consider noise-free retrievals, i.e., no noise is added to the generated synthetic measurements. In this way we focus the experiment on errors related to inconsistencies between the synthetic measurement and retrieval (i.e. different modes), and the capability of the retrieval algorithm itself (for consistent retrievals).

### 4.2 AOT

The synthetic retrievals for AOT are firstly evaluated. The abbreviations for different retrieval cases are summarized in Table 3. For synthetic retrievals, $\chi_{\mathrm{max}}^2 = 0.5$ is chosen as the threshold for $\chi^2$ to define the successfully converged retrievals. Both consistent retrievals and inconsistent retrievals are tested. Consistent retrievals are retrievals for which the mode number for retrievals equals the mode number for creating synthetic measurements. Inconsistent retrievals are the cases when both mode numbers are not equal. Here, although the synthetic measurements do not contain noise, we use the values assumed in the retrieval procedure to compute the $\chi^2$. Note that for consistent retrievals in principle the $\chi^2$ should be much smaller than 0.5 and should even be very close to 0 when the global minimum has been reached. This does obviously not hold for inconsistent retrievals where a different number of modes has been used in the retrieval than in the creation of the synthetic measurements.

Figure 1 shows synthetic retrievals of AOT with the parametric 2-mode retrieval (2modeRetr$^{\mathrm{p}}$) and the 10-mode retrieval (10modeRetr). Both consistent retrievals (i.e., Figure 1a and 1d) and inconsistent retrievals (i.e., Figure 1b and 1c) are performed. To quantitatively evaluate the performances of different retrieval cases, RMSE and bias are indicated. For a fair comparison, RMSE and bias should be calculated for the same number of points. Thus a constant number $n_{\mathrm{validate}}$ ($n_{\mathrm{validate}} < n_{\mathrm{pass}}$) of points are selected to calculate RMSE and bias. In each retrieval case, the selected $n_{\mathrm{validate}}$ ($n_{\mathrm{validate}}$ is chosen at 150 here) points correspond to the points with the smallest $n_{\mathrm{validate}}$ number of $\chi^2$. The total number of retrievals is $n_{\mathrm{pix}}$ ($n_{\mathrm{pix}} = 200$ here).

We first look at the performance of the consistent 10-mode synthetic retrieval, which is shown in Figure 1d. The case is named 10modeRetr+10modeSyn. It shows that the retrieved AOT matches very well with the true AOT. The retrievals at all pixels can pass the strict filter $\chi^2 < 0.5$. Another consistent retrieval case, i.e., parametric 2-mode retrieval on 2-mode synthetic measurements (2modeRetr$^p$+2modeSyn$^p$) is shown in Figure 1a, where the AOT retrieval for $r_{pass}$ (i.e., 98.5%) pixels is also very accurate. Figure 1a and Figure 1d show that both the 10-mode and the parametric 2-mode retrievals have good capabilities of retrieving AOT on consistent synthetic measurements.

Besides consistent retrievals, it is interesting to test the performances of inconsistent retrievals of AOT. This is because in reality, it is unknown how many modes the true atmosphere contains. For this purpose, also inconsistent retrievals are shown: parametric 2-mode retrieval on 10-mode synthetic measurements (2modeRetr$^p$+10modeSyn) in Figure 1b, and 10-mode retrieval on 2-mode synthetic measurements (10modeRetr+2modeSyn$^p$) in Figure 1c. Although AOT retrievals in both inconsistent cases are not as good as those in consistent cases, there is still a good agreement between the retrieved total AOT and the true total AOT over different mode numbers. This shows that inconsistent retrievals are also capable of retrieving AOT.

Next, we check the performances of other multi-mode (i.e., 2-,3-,$\cdots$,9-mode) retrievals. Figure 2 shows the RMSE and the bias for all retrieval cases in the synthetic tests. The x-axis in each subplot represents the parametric 2-mode retrieval (2modeRetr$^p$) and different multi-mode retrieval cases (i.e., 2modeRetr, 3modeRetr, $\cdots$, 10modeRetr). Figure 2a and 2c are for the cases on the 2-mode measurements. It confirms that the parametric 2-mode retrieval as the consistent case has the smallest RMSE and the smallest absolute bias (i.e., closest to zero) compared to inconsistent retrieval cases. Figure 2b and 2d show the cases on the 10-mode measurements. It can be found that the inconsistent retrieval for which $5 < n_{mode} < 10$ have as good performances as the consistent retrieval (10modeRetr+10modeSyn) does. Actually although 3, 4, and 5-mode retrievals on the 10-mode measurements works a bit worse than the multi-mode retrievals with $n_{mode} > 5$, their accuracy is better than the parametric 2-mode retrieval on the 10-mode synthetic measurements. Therefore, we can conclude that multi-mode retrievals have more freedom to be compatible with inconsistent multi-mode measurements. On the other hand, for inconsistent retrievals on 2-mode synthetic measurements, the biases are larger than for the parametric 2-mode retrieval on the 10-mode measurements.

## 4.3   AOT of the fine and coarse modes

It has been investigated that the multi-mode retrievals are capable of retrieving AOT (the total AOT over all modes) for both consistent and inconsistent cases. Since each retrieval case and each measurement case include two types of modes (i.e., the "fine" and "coarse" types), it is interesting to test multi-mode retrievals on the AOT over all fine modes ($\tau_{550}^f$) and the AOT over all coarse modes ($\tau_{550}^c$).

Figure 3 shows $\tau_{550}^f$. For consistent cases (Figure 3a and 3d), the retrievals are accurate and nearly unbiased. For inconsistent cases (Figure 3b and 3c), there are clear underestimations. This generally happens in inconsistent retrievals on the 2-mode measurements, which can be seen in Figure 3g (where all the inconsistent retrievals show a negative bias). This doesn't happen for inconsistent retrievals on the 10-mode measurements, where the parametric 2-mode retrieval (2modeRetr$^p$), the fixed 2-mode retrieval (2modeRetr), and the 3-mode retrieval (3modeRetr) underestimate $\tau_{550}^f$; the 4-mode retrieval (4modeRetr) and the 5-mode retrieval (5modeRetr) slightly overestimate $\tau_{550}^f$; retrievals are almost unbiased for $\tau_{550}^f$ if $n_{mode} > 5$. By checking

the RMSE of all retrieval cases on the 2-mode measurements (Figure 3e) and the RMSE on the 10-mode measurements (Figure 3f), retrievals have quite acceptable accuracies on both types of measurements if $n_{\mathrm{mode}} > 3$.

The total AOT of the coarse modes ($\tau_{550}^{\mathrm{c}}$) is shown in Figure 4. Compared to the underestimation in Figure 3b and 3c for $\tau_{550}^{\mathrm{f}}$, there is an overestimation for $\tau_{550}^{\mathrm{c}}$, as shown in Figure 4b and 4c. The reverse bias between $\tau_{550}^{\mathrm{f}}$ and $\tau_{550}^{\mathrm{c}}$ results in total AOT over all modes that is almost unbiased, as shown in Figure 1. These offset effects can also be seen by comparing Figure 4g and 3g or by comparing Figure 4h and 3h. According to the RMSE shown in Figure 4e and 4f, retrievals with $n_{\mathrm{mode}} > 3$ have good retrieval accuracy of $\tau_{550}^{\mathrm{c}}$ on both synthetic measurements.

**4.4 SSA**

We also tested multi-mode retrievals of Single Scattering Albedo (SSA). Figure 5 shows the parametric 2-mode retrieval and the 10-mode retrieval for SSA, while Figure 6 shows the RMSE and the bias for different retrieval cases for the difference between the retrieved SSA and the true SSA.

   By comparing SSA for consistent retrieval cases (Figure 5a and 5d), for the $n_{\mathrm{pass}}$ pixels (marked as "red" points), the match
between the retrieved SSA and the true SSA in Figure 5a is slightly worse than the match in Figure 5d. This demonstrates the challenge in retrieving $r_{\mathrm{eff}}$ and $v_{\mathrm{eff}}$ in the parametric 2 mode approach – even for a consistent setup – since $r_{\mathrm{eff}}$ and $v_{\mathrm{eff}}$ are affecting the derived SSA. For inconsistent retrievals however, we see that the parametric 2-mode retrieval on the 10-mode synthetic measurements works better than vice versa. Although Figure 5 shows different performances among the parametric 2-mode retrieval (2modeRetr$^{\mathrm{p}}$) and the 10-mode retrieval (10modeRetr), the accuracy and the bias in the four cases are quite
good.

   Figure 6b and 6d show the RMSE and bias comparisons among all retrieval cases on the 10-mode synthetic measurements. All retrievals for SSA except the fixed 2-mode case are shown to be accurate and have small bias. On the 2-mode synthetic measurements, the RMSE (Figure 6a) of multi-mode retrievals are a bit worse than the consistent SSA retrieval. For the bias on the 2-mode synthetic measurement shown in Figure 6c, it varies between 0 and -0.005. For the 10-mode synthetic
measurements, the retrievals are virtually unbiased if $n_{\mathrm{mode}} > 4$ and for the parametric 2-mode retrieval.

**4.5 Refractive index**

**4.5.1 Real part of refractive index**

As described in Sect. 2.2, also for multi-mode retrievals we use a separate refractive index for the fine and coarse mode, respectively. In this case, the fine mode refractive index corresponds to mode number 1-6 in Table 2 and the coarse mode
refractive index to mode 7-10. Here we first test the retrievals of the real part of the refractive index for the fine modes and the coarse modes, i.e., $m_{\mathrm{r}}^{\mathrm{f}}$ and $m_{\mathrm{r}}^{\mathrm{c}}$ (at wavelength 550 nm), as respectively shown in Figure 7 and 8.

   For the consistent retrievals (2modeRetr$^{\mathrm{p}}$+2modeSyn$^{\mathrm{p}}$ and 10modeRetr+10modeRetr), $m_{\mathrm{r}}^{\mathrm{f}}$ is retrieved with small RMSE and nearly unbiased, as shown in Figure 7a and 7d. Similarly, $m_{\mathrm{r}}^{\mathrm{c}}$ is also well retrieved in the consistent retrievals, which

are shown in Figure 8a and 8d. Actually, $m_r^c$ retrieval is shown better than $m_r^f$ retrieval, and 10-mode retrieval on 10-mode synthetic measurements is shown better than parametric 2-mode retrieval on 2-mode synthetic measurements.

For inconsistent retrieval cases, we first check the performances on the 10-mode measurements, i.e., the right panel of Figure 7 and 8. It shows that the parametric 2-mode retrieval and the multi-mode retrievals with $n_{mode} > 4$ are capable of retrieving $m_r^f$ and $m_r^c$. However, this is not the case for retrievals on the 2-mode measurements, i.e., the left panel in Figure 7 and 8. $m_r^f$ is retrieved with overestimation, as shown in Figure 7b and 7g. For the retrieval of $m_r^c$ (see Figure 8b and 8g) an underestimation can be observed. It can be concluded that the parametric 2-mode retrieval works better for fine mode real part

of refractive index than the multi-mode retrievals.

### 4.5.2 Imaginary part of refractive index

Next, we test the retrievals of the imaginary part of the refractive index. The fine mode and coarse mode cases (i.e., $m_i^f$ and $m_i^c$) are respectively shown in Figure 9 and 10.

For consistent retrievals, $m_i^f$ is shown to be well retrieved for both the parametric 2-mode case and the 10-mode case, see

Figure 9a and 9d. This is a similar result with the consistent retrievals of $m_r^f$ (Figure 7a and 7d). However, for the coarse mode case, the consistent retrievals of $m_i^c$ (Figure 10a and 10d) do not look as good as the consistent retrievals of $m_r^c$ (Figure 8a and 8d), especially for the consistent parametric 2-mode case (Figure 10a), where there are some clear outliers. Based on these results, we conclude that for the consistent cases, (1) $m_i^f$ retrieval is better than $m_i^c$ retrieval; (2) 10-mode retrieval of $m_i^f$ and $m_i^c$ looks better than the parametric 2-mode retrieval.

For inconsistent retrieval cases, the performances on the 10-mode synthetic measurements (see the right panel of Figure 9 and 10) show that $m_i^f$ and $m_i^c$ can be well retrieved in the parametric 2-mode and the multi-mode retrievals with $n_{mode} > 4$. This result is similar to what was shown for the inconsistent retrievals of $m_r^f$ and $m_r^c$ (see the right panel of Figure 7 and 8), except for one difference, i.e., the parametric 2-mode retrieval has clear overestimation when retrieving $m_i^f$, as shown in Figure 9c or Figure 9h. Now we check inconsistent retrievals on the 2-mode measurements. For $m_i^f$ (see the left panel of Figure 9), clear

overestimation can be observed. For $m_i^c$ (see the left panel of Figure 10), the multi-mode retrievals with $n_{mode} > 4$ are quite accurate and only slightly underestimate $m_i^c$. We can therefore conclude that the multi-mode retrievals with $n_{mode} > 4$ work slightly better than the parametric 2-mode retrieval for fine and coarse mode imaginary part of refractive index.

### 4.6 Height

The retrievals of the central height $z$ of the aerosol layer are shown in Figure 11. It can be seen that $z$ can be well retrieved in

the consistent retrievals (RMSE $< 50$ m, bias $\approx$ -5 m), as shown in Figure 11a and 11d.

For the inconsistent retrievals on the 2-mode synthetic measurements, $4, 5, 6, 7, 9, 10$-mode retrievals (RMSE $\approx 200$ m, bias $\approx$ -200 m) perform better than other inconsistent cases, which are shown in Figure 11b, 11e and 11g. For inconsistent retrievals on the 10-mode synthetic measurements, parametric 2-mode retrieval performs with clear underestimation as shown in Figure 11c (bias $= -396.2$ m), but multi-mode retrievals with $n_{mode} > 5$ perform very well with high accuracy (RMSE $< 20$ m)

and little bias, as shown in Figure 11f and 11g. To summarize, for inconsistent retrievals, the RMSE is typically around 500 m and the bias is around 300 m.

Based on results above, we conclude that the multi-mode retrievals with $n_{\mathrm{mode}} > 5$ are capable of retrieving the central height of the aerosol layer.

### 4.7   Pass rate of synthetic retrievals

The pass rate $r_{\mathrm{pass}}(\chi^2 < \chi^2_{\mathrm{max}}, \chi^2_{\mathrm{max}} = 0.5)$ for the parametric 2-mode retrieval (2modeRetr$^{\mathrm{p}}$) and the multi-mode retrievals are shown in Figure 12. In both Figure 12a and 12b, the fixed 2-mode retrieval (2modeRetr) has the smallest pass rate

($r_{\mathrm{pass}} \approx 55\%$) compared to the other retrieval cases. This is an indication that 2 fixed modes are not enough. Figure 12a shows retrievals on 2-mode measurements (2modeSyn$^{\mathrm{p}}$). The $r_{\mathrm{pass}}$ for retrievals with $n_{\mathrm{mode}} > 2$ are about 75% to 90%. The highest pass rate (up to 98.5%) in Figure 12a is reached by the parametric 2-mode retrieval on 2-mode synthetic measurements. Figure 12b shows the retrievals on 10-mode measurements (10modeSyn). The pass rates are high (95% to 100%) for all retrieval cases except for the 2 fixed mode retrieval.

## 15  5   Real data retrievals

### 5.1   Experimental setup

The synthetic experiments above have shown multi-mode retrievals with $n_{\mathrm{mode}} > 5$ have the capability to retrieve aerosol optical and microphysical properties. Next, we test the performances of multi-mode retrievals on real data, i.e., PARASOL satellite data, as introduced in Sect. 3.1.

To validate PARASOL (satellite) retrievals, AERONET (ground-based) AOT and SSA data are used, as introduced in Sect. 3.3. AERONET measurements at 20 stations (listed in Table 7) in the year 2006 are used in this study to validate multi-mode retrievals on real data. To make PARASOL retrievals and AERONET data comparable, only the PARASOL retrievals within 20 km around each AERONET station are selected. The AERONET data are averaged within 2 h (h = hours) from PARASOL.

### 25  5.2   AOT: multi-mode retrievals versus parametric 2-mode retrieval

In this section, the performances of multi-mode retrievals for AOT are compared to that of the parametric 2-mode retrieval. Figure 13 shows real data retrievals of AOT among the parametric 2-mode retrieval the 5-mode retrieval and the 10-mode retrieval at three different wavelengths, i.e., 440 nm, 675 nm and 870 nm, which are respectively represented by the three columns in Figure 13.

We first focus on the performances at 675 nm, i.e., Figure 13b (2modeRetr$^{\mathrm{p}}$), 13e (5modeRetr) and 13h (10modeRetr). The total number of PARASOL retrievals for the 20 AERONET stations is $n_{\mathrm{pix}}$ ($n_{\mathrm{pix}} = 63488$ here). For real data retrievals, $\chi^2_{\mathrm{max}} = 5.0$ is used as the filter for goodness of fit. The total number of pixels where the retrieval passes $\chi^2 < \chi^2_{\mathrm{max}}$ is $n_{\mathrm{pass}}$.

The number of red points shown in each figure is not $n_{\mathrm{pass}}$, but $n_{\mathrm{averaged}}$ ($n_{\mathrm{averaged}} \approx 1100$ here), which represents the number of $n_{\mathrm{pass}}$ retrievals after daily averages. The magenta points represent the $n_{\mathrm{validate}}$ ($n_{\mathrm{validate}} = 1000$ here) best retrievals corresponding to the smallest $\chi^2$, which is needed if we want to compare the different retrieval setups for the same number of measurements.

For real data retrievals, we set $\chi^2_{\mathrm{max}}$ at 5.0, which means that we actually underestimated the assumed errors in the retrieval (otherwise the $\chi^2$ would be around 1.0). The pass rates for the parametric 2-mode retrieval and the multi-mode retrievals are between 32.5% and 40.8%. Comparing Figure 13b, 13e and 13h, the 10-mode retrieval performs the best with the smallest RMSE (0.1230) and the smallest absolute bias (0.0048). While, the parametric 2-mode retrieval has the largest RMSE (0.1624) and the 5-mode retrieval has the largest absolute bias (0.0301).

Besides these three retrieval cases, we also perform multi-mode retrievals with different number of modes. The RMSE and the bias for all the retrieval cases (2 – 10 modes) are shown in Figure 14. From Figure 14a, it is seen that multi-mode retrievals generally have better agreement with AERONET than parametric 2-mode retrieval, especially for the multi-mode retrievals with $n_{\mathrm{mode}} > 4$. From Figure 14b, it can be found that the parametric 2-mode retrieval has an overestimation (0.019) and all the multi-mode retrievals show an underestimation. The 10-mode retrieval is almost unbiased, with the smallest underestimation. Other multi-mode retrievals show larger underestimation (from 0.0235 to 0.0429).

Based on the results above, we can conclude that multi-mode retrievals generally work better for retrieving AOT than the parametric 2-mode retrieval. However, multi-mode (except for 10-mode) retrievals have larger absolute bias than the parametric 2-mode retrieval.

## 5.3   AOT: multi-mode retrievals for different wavelengths

Sect. 5.2 discussed the retrieval performances at 675 nm. It is interesting to see how the retrievals perform at other wavelengths. For this purpose, 440 nm and 870 nm are chosen to evaluate the results.

We compare the three sub-figures in each row of Figure 13, e.g., Figure 13g (440 nm), 13h (675 nm) and 13i (870 nm). It can be observed that for the 10-mode retrieval (10modeRetr) at 440 nm, 675 nm and 870 nm, the RMSE are respectively 0.1654, 0.1230 and 0.1188. For the 5-mode retrieval (5modeRetr), the RMSE are respectively 0.2152, 0.1513 and 0.1305. For the parametric 2-mode retrieval (2modeRetr[p]), the RMSE are respectively 0.2209, 0.1624 and 0.1403. It can be therefore found that as the wavelength increases, the retrieval accuracy improves in absolute sense. However, this is mainly caused by the fact that the AOT value itself decreases with wavelength.

Second, we check at 440 nm and 875 nm, whether the conclusions at 675 nm hold. For this purpose, we look at the first and the third column of Figure 13. It can be seen that the RMSE decreases going from the parametric 2-mode retrieval, the 5-mode retrieval, to the 10-mode retrieval. This means that the retrieval accuracy at 440 nm (or 875 nm) improves as the mode number increases. Therefore, the conclusion at 675 nm also holds for other wavelengths.

## 5.4 SSA

Next we validate PARASOL retrievals of Single Scattering Albedo (SSA) with the AERONET-based SSA (described in Sect. 3.3). The AERONET SSA itself is not a result from a direct measurement but from an inversion procedure with different kind of assumptions (Dubovik et al., 2002). The error in the AERONET SSA is at least 0.03 (Dubovik et al., 2002). The comparisons shown in this section should be interpreted taking this uncertainty into account.

Similarly to what was shown for AOT (Figure 13), Figure 15 shows SSA comparisons for the same retrieval setups as above (2modeRetr[P], 5modeRetr and 10modeRetr) at 440 nm, 675 nm and 870 nm. For SSA, it is usually difficult to retrieve it when AOT is small, thus shown in Figure 15 are the SSA retrievals when AOT is larger than 0.3 at the corresponding wavelength.

We first check the tendency of the SSA accuracy for different wavelengths. By comparing RMSE in each row of Figure 15, it can be found that the RMSE increases as the wavelength increases for all setups. Thus, PARASOL retrievals of SSA have an "accuracy decreasing" tendency as the wavelength increases. The reason is (again) that the AOT decreases with wavelength and the SSA retrieval becomes less accurate for decreasing AOT. The reverse is true for AOT retrievals as discussed in Sect. 5.3. Note that for the parametric 2-mode retrieval, the RMSE at 675 nm (0.0601) in Figure 15 is actually smaller than the RMSE at 440 nm (0.0629), but the difference is small.

Comparing RMSE in each column of Figure 15, it can hardly be concluded which one among the different retrieval setups (2modeRetr[P], 5modeRetr and 10modeRetr) compares best against AERONET. For example, the 10-mode retrieval performs better at 440 nm and 675 nm, but the parametric 2-mode retrieval performs better at 870 nm. Different retrieval setups for SSA seem to have similar accuracies. This can be confirmed by Figure 16a, where RMSE values are varying within a small interval (0.0577 to 0.0611) for most retrieval cases except for the fixed 2-mode retrieval, the 3-mode retrieval and the 5-mode retrieval. As for the bias (Figure 16b), all the setups show an overestimation and the bias values in all the retrieval cases are quite similar (except for the fixed 2-mode retrieval). Based on the comparison above, we can conclude that multi-mode retrievals have similar performances as the parametric 2-mode retrieval for SSA.

For the PARASOL retrievals in this paper we did not retrieve the aerosol layer height but used a fixed value of 1 km. This resulted in better AOT retrievals. The reason for poor performance of aerosol height retrieval from PARASOL is probably caused by the absence of of near-UV polarization measurements in combination with the relatively poor polarimetric accuracy (Wu et al., 2016).

## 6 Discussions and conclusions

In this study we compared aerosol retrievals from Multi-Angle Polarimeter (MAP) data for different definitions of the retrieval state vector: (1) a 2-mode definition where the state vector includes aerosol properties for fine/coarse modes and land or ocean surface properties; (2) a multi-mode definition where the state vector excludes the effective radius and the effective variance and only retrieves the aerosol column of each mode. For the purpose of this study we extended the SRON-Aerosol algorithm – which was based a parametric 2-mode approach – to include capability of a multi-mode retrieval. To evaluate the retrieval

capability for different state vector definitions, the performances between multi-mode approaches and the parametric 2-mode retrieval approach were compared on both synthetic measurements and real (PARASOL) measurements.

5    In synthetic experiments, the consistent retrievals (when the number of modes for retrievals equals to the number of modes for creating synthetic measurements) show both the multi-mode and parametric 2-mode approach can reach high accuracy for most of the parameters, e.g., the Aerosol Optical Thickness (AOT), the Single Scattering Albedo (SSA), the refractive index, the aerosol height. For inconsistent retrievals on 10-mode synthetic measurements, the multi-mode retrievals with $n_{\mathrm{mode}} > 5$ were shown to be capable of retrieving aerosol properties with sufficient accuracy, and they perform similar as the parametric 2-mode retrievals. The good performances of multi-mode approaches indicate that multi-mode retrievals have good compatibility with different kinds of measurements.

It should be noted that the geometry used for the synthetic study in this paper is quite favorable as it assumes measurements in the principal plane. We also did the same synthetic study for a much less favorable geometry (SZA=20°, relative azimuth angle=60°/-120°). Although for the latter geometry, the performance is somewhat worse, the main conclusions from
the synthetic study still hold for this geometry.

After synthetic experiments, real (PARASOL) data experiments were performed. Multi-mode retrievals of AOT were shown to compare better to AERONET than the parametric 2-mode retrieval (e.g., RMSE 0.1230 over 0.1624). Here, we found that the agreement with AERONET improves with increasing number of modes, with the 10 mode retrieval showing the best agreement with AERONET for AOT. For real data retrievals of SSA, both multi-mode and parametric 2-mode retrievals have
similar performances.

When comparing retrievals between different algorithms, it is important to realize that the performance of a given algorithm depends on a number of factors, the definition of the aerosol state vector being one of them. Other factors are the inversion approach (cost function, regularization strength, multiple versus single pixel), the accuracy of the forward model, and the surface reflection model. It is important to study the above mentioned aspects with an individual algorithm. However, now that
the SRON algorithm has been extended to include an arbitrary number of fixed modes, it has become easier to compare to other algorithms using a similar state vector definition (Dubovik et al., 2011; Xu et al., 2017). This would be an important topic for future research.

The multi-mode approach provides an opportunity to make aerosol retrievals more computationally efficient. This is due to the fact that the effective radius and the effective variance are not retrieved in the multi-mode retrievals, thus the Mie/T-Matrix
calculation for each mode can be fixed and pre-computed as function of refractive index. Then, there is no need to integrate over size distribution during the retrieval. Therefore, the most time-consuming part (as it is called many times) of the retrieval can be significantly accelerated.

*Data availability.* The PARASOL level-1 data can be downloaded from the website: http://www.icare.univ-lille1.fr/parasol/products. The AERONET data can be downloaded from the website: https://aeronet.gsfc.nasa.gov/. The meteorological NCEP data can be accessed through the website: http://www.cdc.noaa.gov/. The retrieval results will be made available on SRON's ftp site.

*Competing interests.* The authors declare that no competing interests are present.

*Acknowledgements.* This work is funded by a NWO/NSO project ACEPOL: Aerosol Characterization from Polarimeter and Lidar under
5   project number ALW-GO/16-09. We thank PARASOL team and AERONET team to maintain the data. NCEP Reanalysis data provided by
the NOAA/OAR/ESRL PSD, Boulder, Colorado, USA, from their website at https://www.esrl.noaa.gov/psd/. We would also like to thank
the Netherlands Supercomputing Centre (SURFsara) for providing us with the computing facility, the Cartesius cluster. We are very grateful
to the editor, Dr Mishchenko and Dr Lang for their reviews and insightful comments.

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

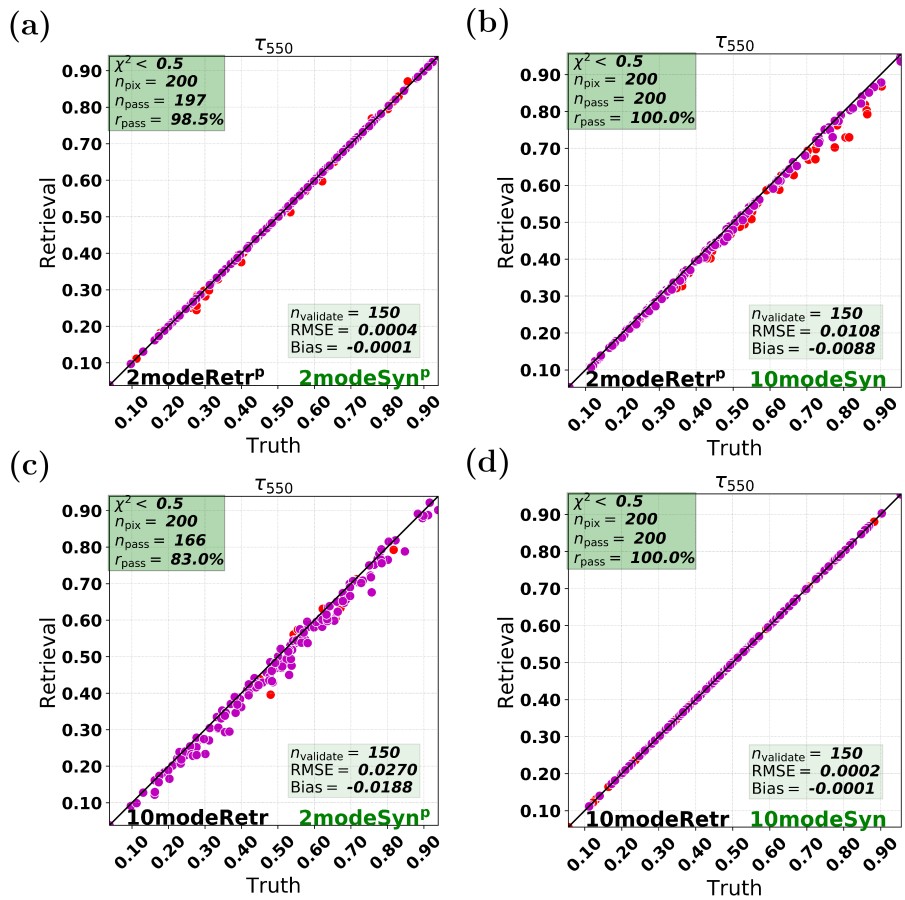

**Figure 1. Synthetic retrievals: Aerosol Optical Thickness (AOT) with the parametric 2-mode retrieval (2modeRetr[P]) and the 10-mode retrieval (10modeRetr).** The red and magenta points represent $n_{pass}$ and $n_{validate}$ points, respectively. The measurements are the parametric 2-mode synthetic measurement (2modeSyn[P]) and the 10-mode synthetic measurement (10modeSyn). **(a)** 2modeRetr[P] on 2modeSyn[P]. **(b)** 2modeRetr[P] on 10modeSyn. **(c)** 10modeRetr on 2modeSyn[P]. **(d)** 10modeRetr on 10modeSyn.

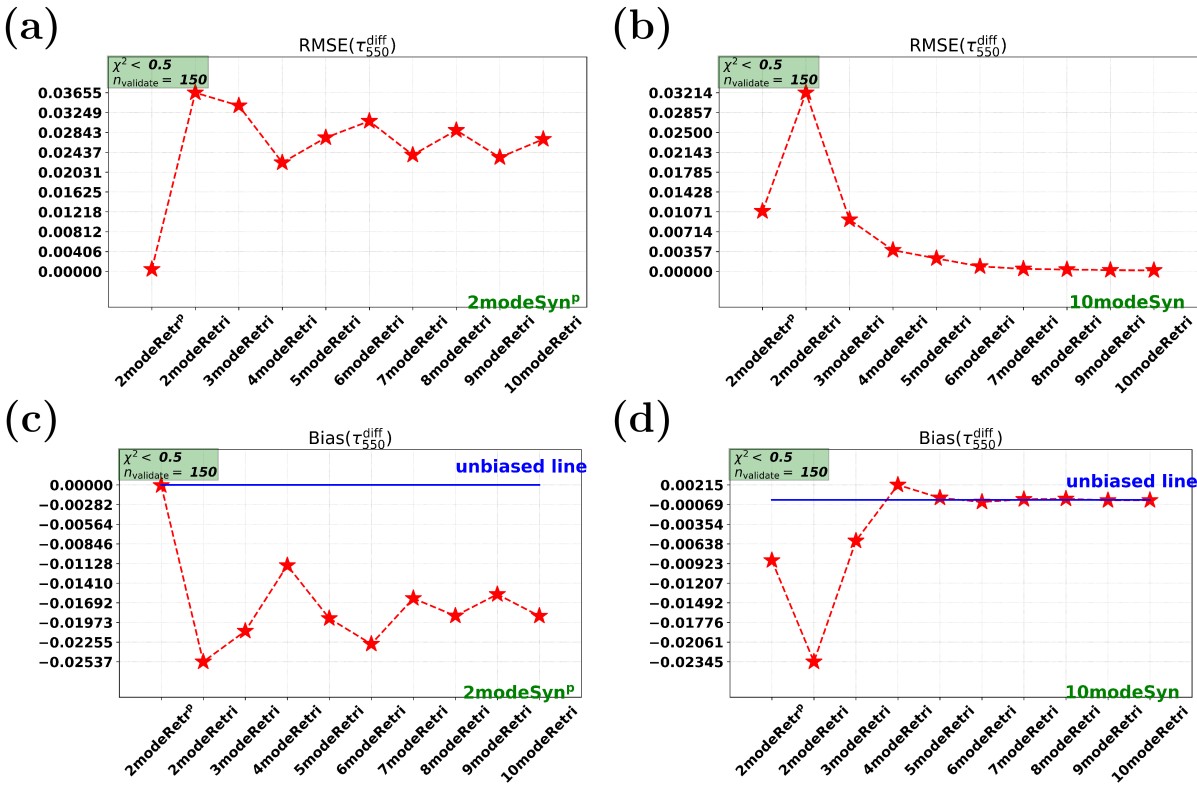

**Figure 2. Synthetic retrievals for AOT: Root-Mean-Square error (RMSE) and bias for the difference between the retrieved AOT and the true AOT.** The x-axis in each subplot represents 2modeRetr$^P$ and different multi-mode retrieval cases (i.e., 2modeRetr, 3modeRetr, $\cdots$, 10modeRetr). **(a), (c)** are RMSE and bias for the cases on 2modeSyn$^P$. **(b), (d)** are RMSE and bias for the cases on 10modeSyn.

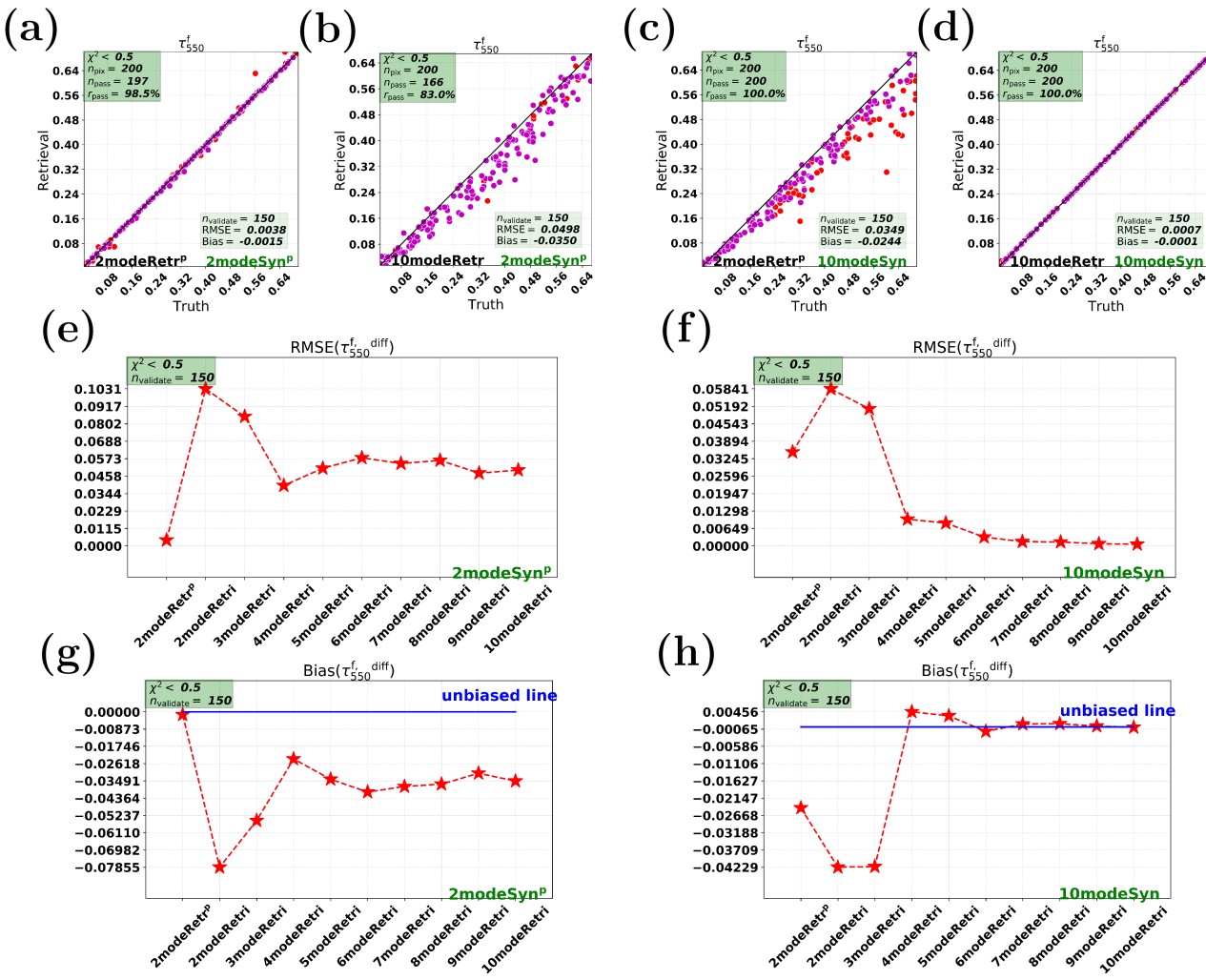

**Figure 3. Synthetic retrievals for the AOT (at 550 nm) of all the fine modes** ($\tau_{550}^{\mathrm{f}}$)**.** The left panel (i.e., **(a), (b), (e), (g)**) and the right panel (i.e., **(c), (d), (f), (h)**) are for cases on 2modeSyn$^{\mathrm{P}}$ and 10modeSyn, respectively. Row 1 shows the parametric 2-mode retrieval (2modeRetr$^{\mathrm{P}}$) and the 10-mode retrieval (10modeRetr). Row 2 shows RMSE and row 3 shows bias for different retrieval cases.

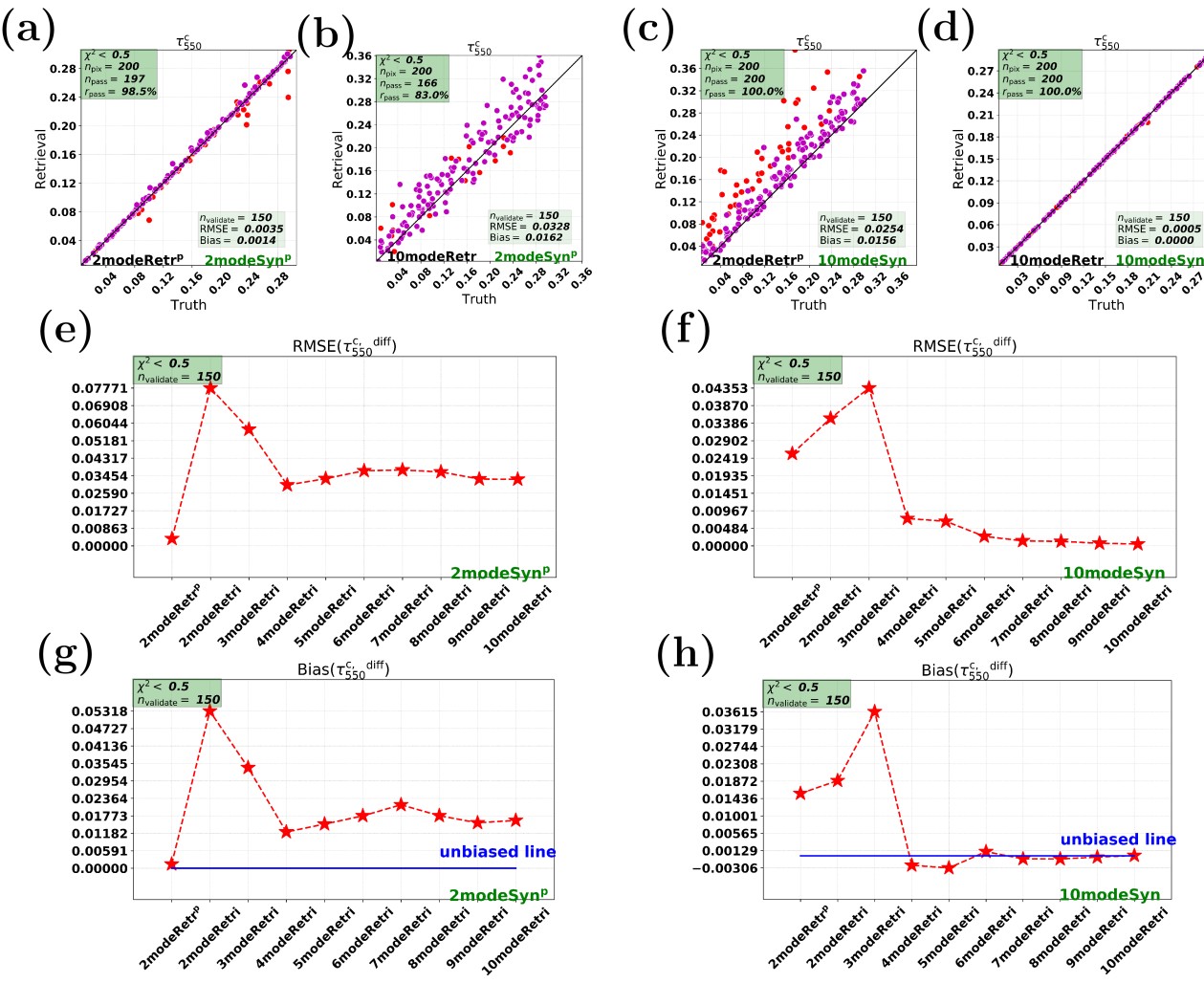

**Figure 4. Synthetic retrievals for the AOT (at 550 nm) of all the coarse modes** ($\tau^c_{550}$). The left panel and the right panel are for cases on 2modeSyn$^P$ and 10modeSyn, respectively.

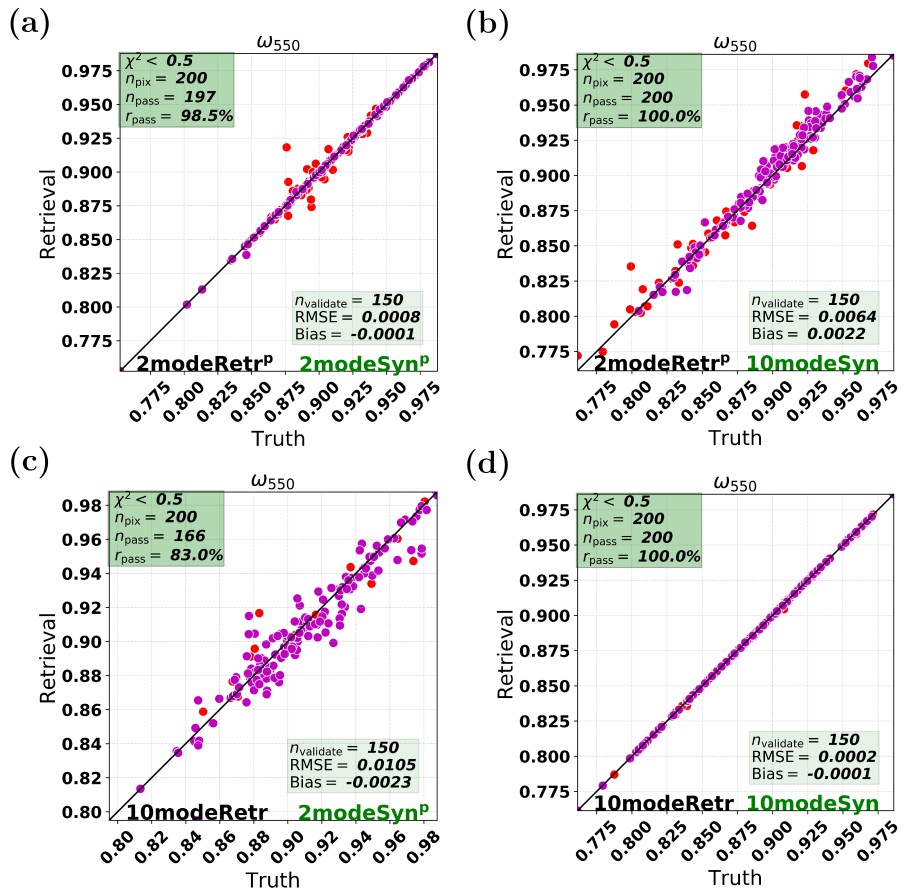

**Figure 5. Synthetic retrievals: Single Scattering Albedo (SSA) with the parametric 2-mode retrieval (2modeRetr$^\mathrm{P}$) and the 10-mode retrieval (10modeRetr).** The red and magenta points represent $n_\mathrm{pass}$ and $n_\mathrm{validate}$ points, respectively. The measurements are the parametric 2-mode synthetic measurement (2modeSyn$^\mathrm{P}$) and the 10-mode synthetic measurement (10modeSyn). **(a)** 2modeRetr$^\mathrm{P}$ on 2modeSyn$^\mathrm{P}$. **(b)** 2modeRetr$^\mathrm{P}$ on 10modeSyn. **(c)** 10modeRetr on 2modeSyn$^\mathrm{P}$. **(d)** 10modeRetr on 10modeSyn.

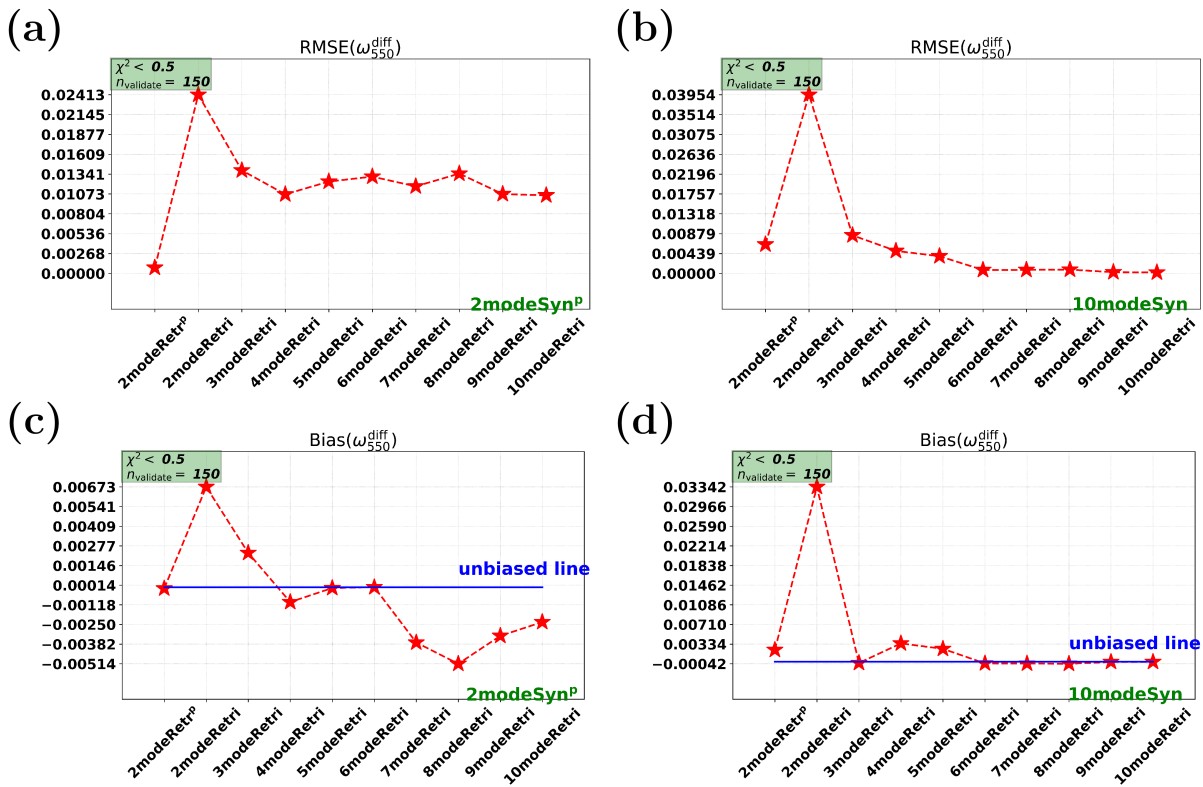

**Figure 6. Synthetic retrievals for SSA: Root-Mean-Square error (RMSE) and bias for the difference between the retrieved SSA and the true SSA. (a), (c)** are RMSE and bias for the cases on 2modeSyn[P]. **(b), (d)** are RMSE and bias for the cases on 10modeSyn.

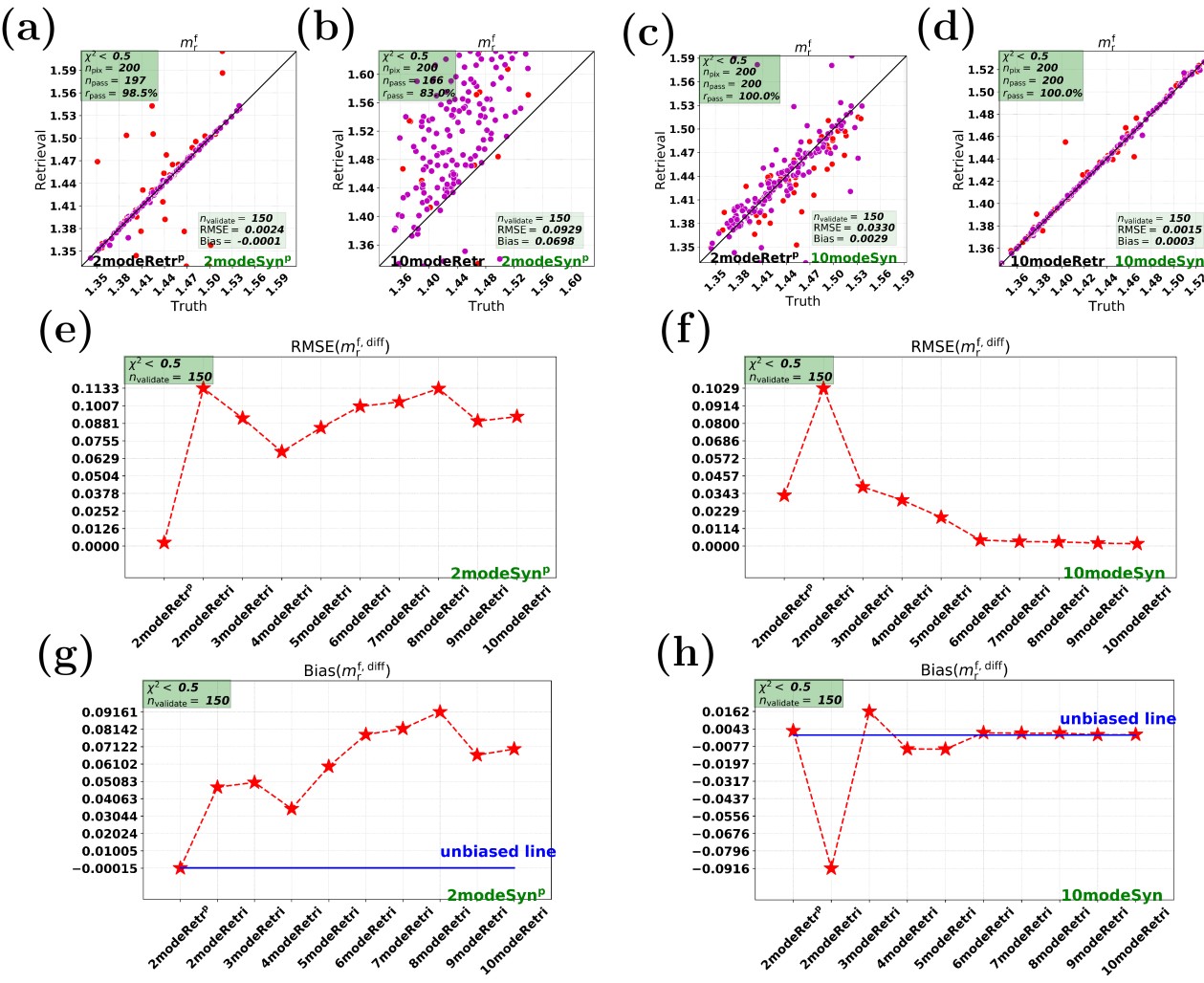

**Figure 7. Synthetic retrievals for the real part of refractive index (at 550 nm) of the fine modes ($m_r^f$).** The left panel and the right panel are for cases on 2modeSyn$^p$ and 10modeSyn, respectively.

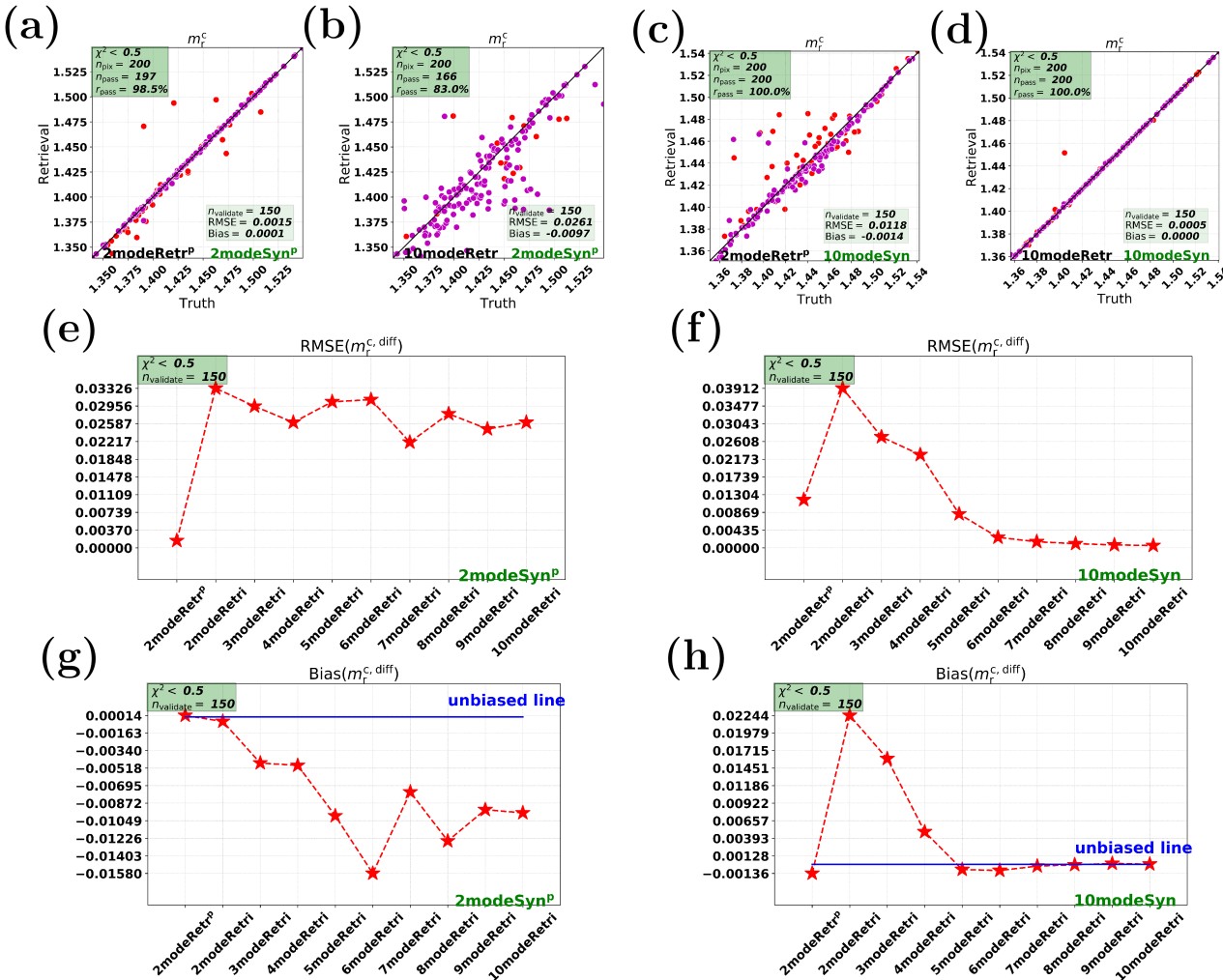

**Figure 8. Synthetic retrievals for the real part of refractive index (at 550 nm) of the coarse modes** ($m_r^c$). The left panel and the right panel are for cases on 2modeSyn[p] and 10modeSyn, respectively.

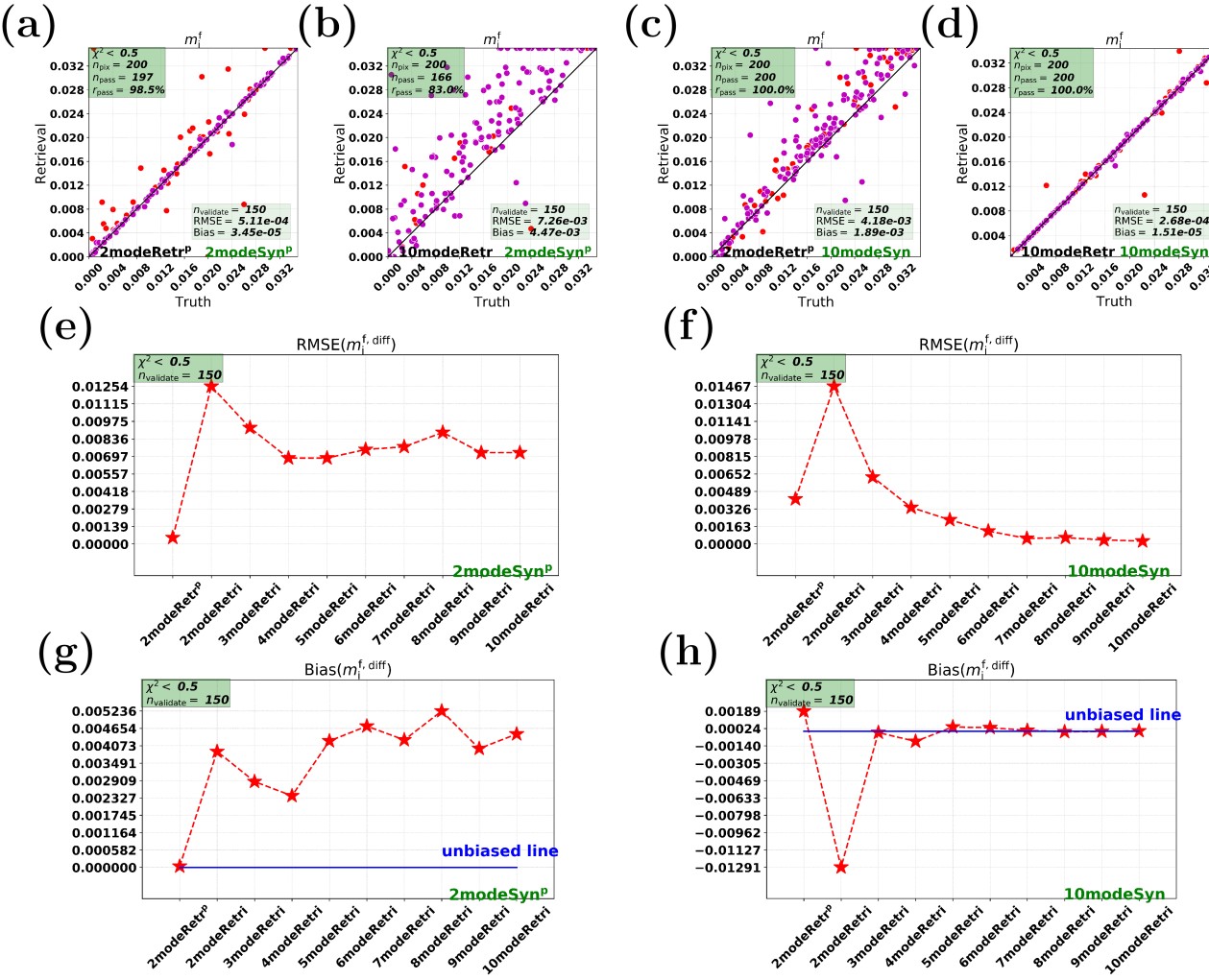

**Figure 9. Synthetic retrievals for the imaginary part of refractive index (at 550 nm) of the fine modes** ($m_i^f$). The left panel and the right panel are for cases on 2modeSyn$^p$ and 10modeSyn, respectively.

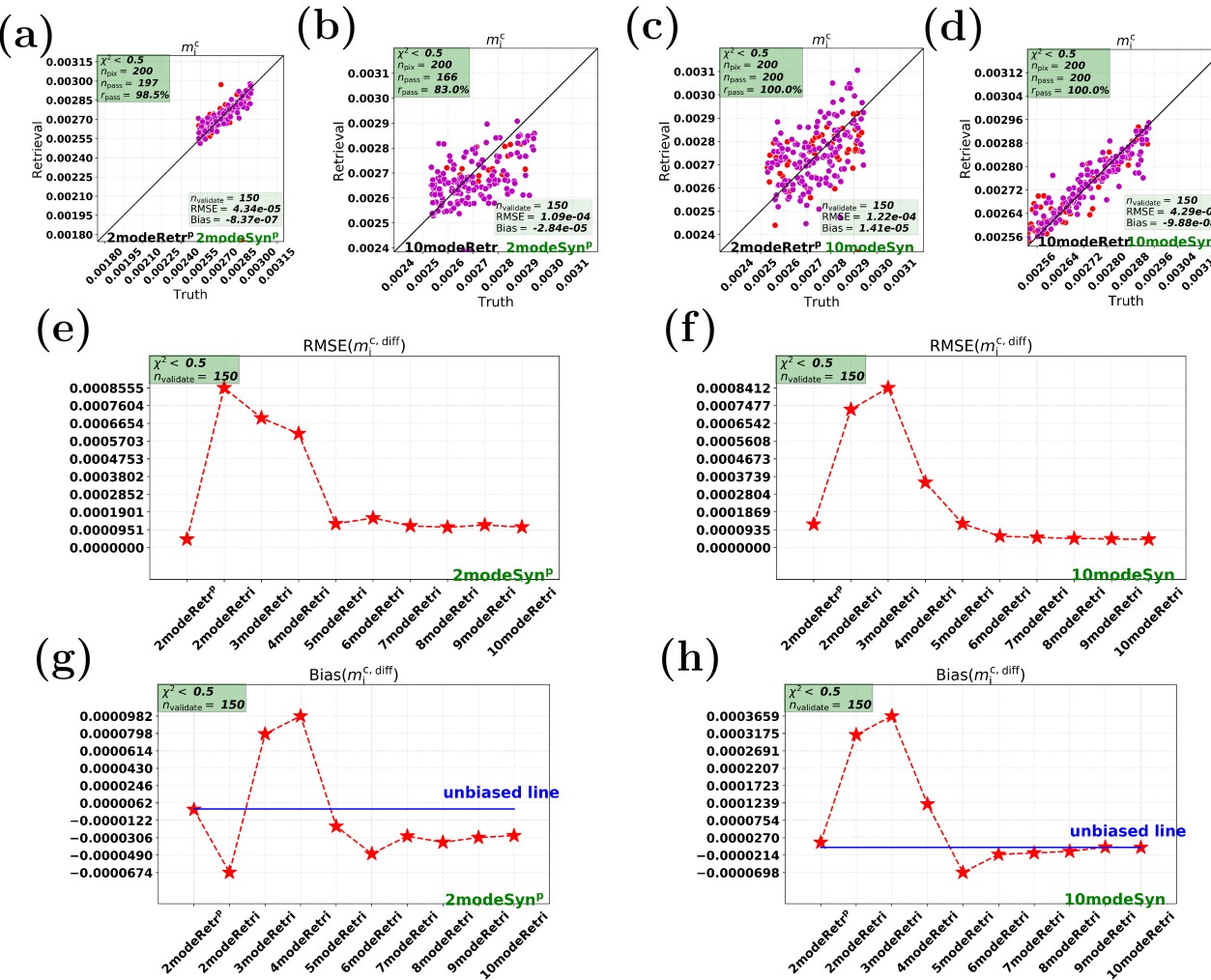

**Figure 10. Synthetic retrievals for the imaginary part of refractive index (at 550 nm) of the coarse modes ($m_i^c$).** The left panel and the right panel are for cases on 2modeSyn$^p$ and 10modeSyn, respectively.

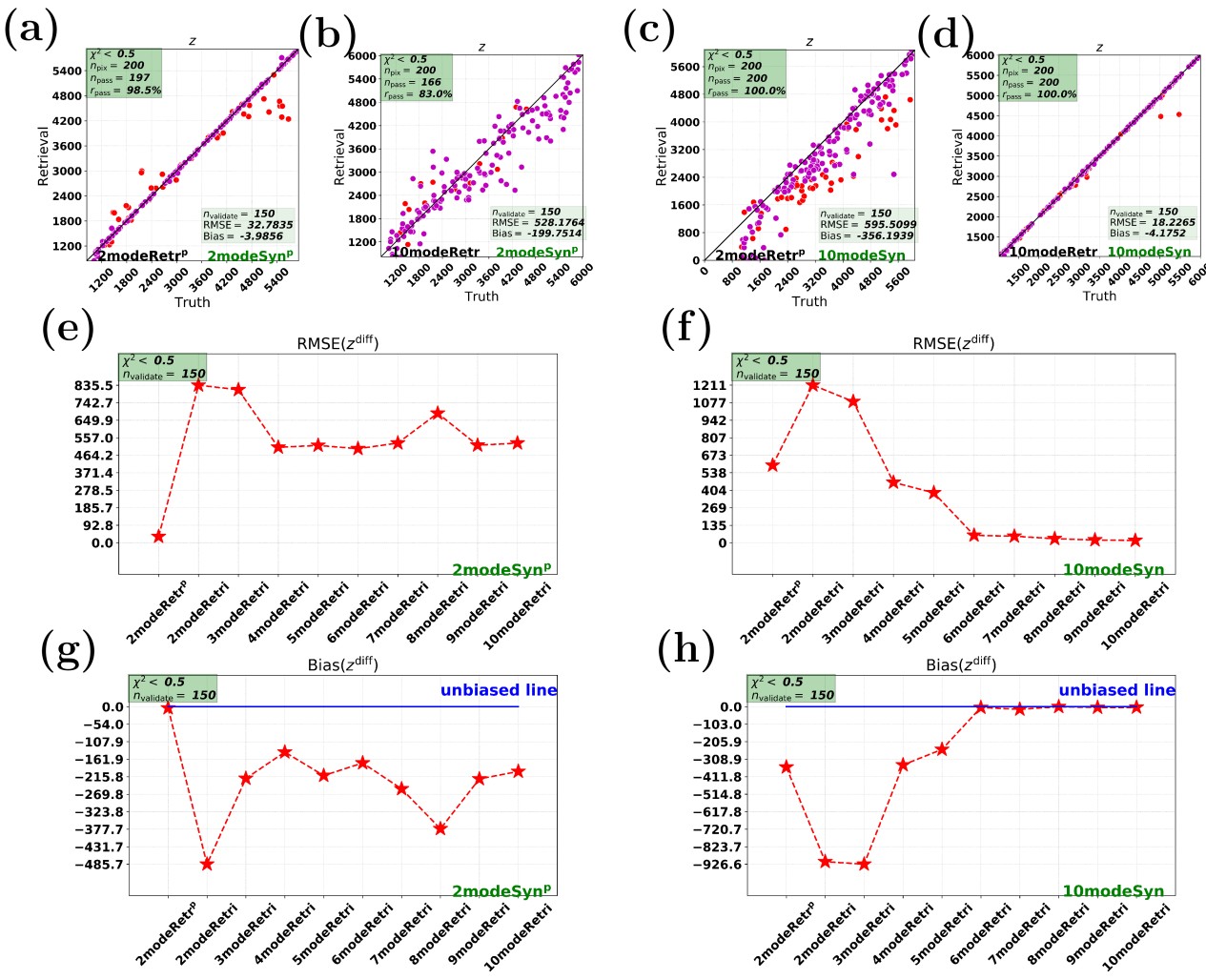

**Figure 11. Synthetic retrievals for the central height ($z$) of the aerosol layer.** The left panel and the right panel are for cases on 2modeSyn$^P$ and 10modeSyn, respectively.

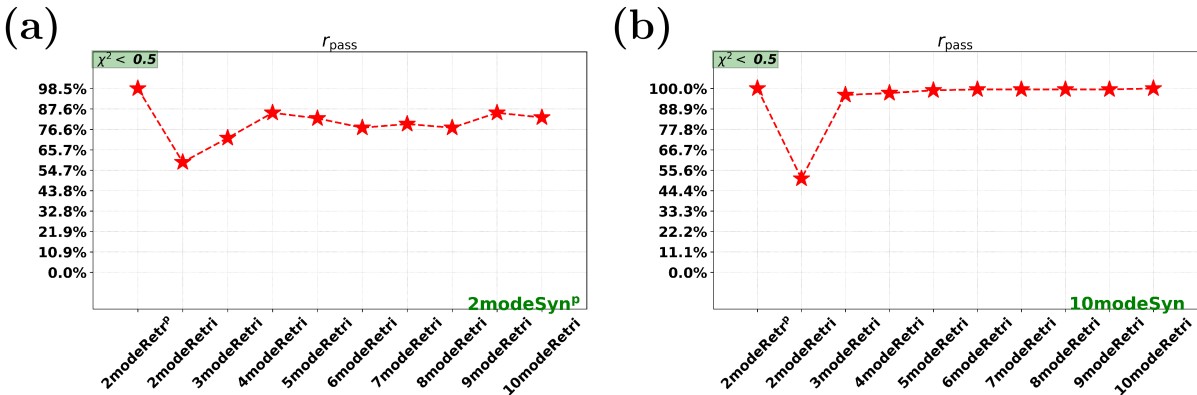

**Figure 12. Synthetic retrievals: pass rates when $\chi^2 < 0.5$.** **(a)** Retrievals on the parametric 2-mode based synthetic measurement (2modeSyn[P]). **(b)** Retrievals on the 10-mode based synthetic measurement (10modeSyn). The x-axis in the subplot represents the parametric 2-mode retrieval (2modeRetr[P]) and different multi-mode retrieval cases (i.e., 2modeRetr, 3modeRetr, $\cdots$, 10modeRetr).

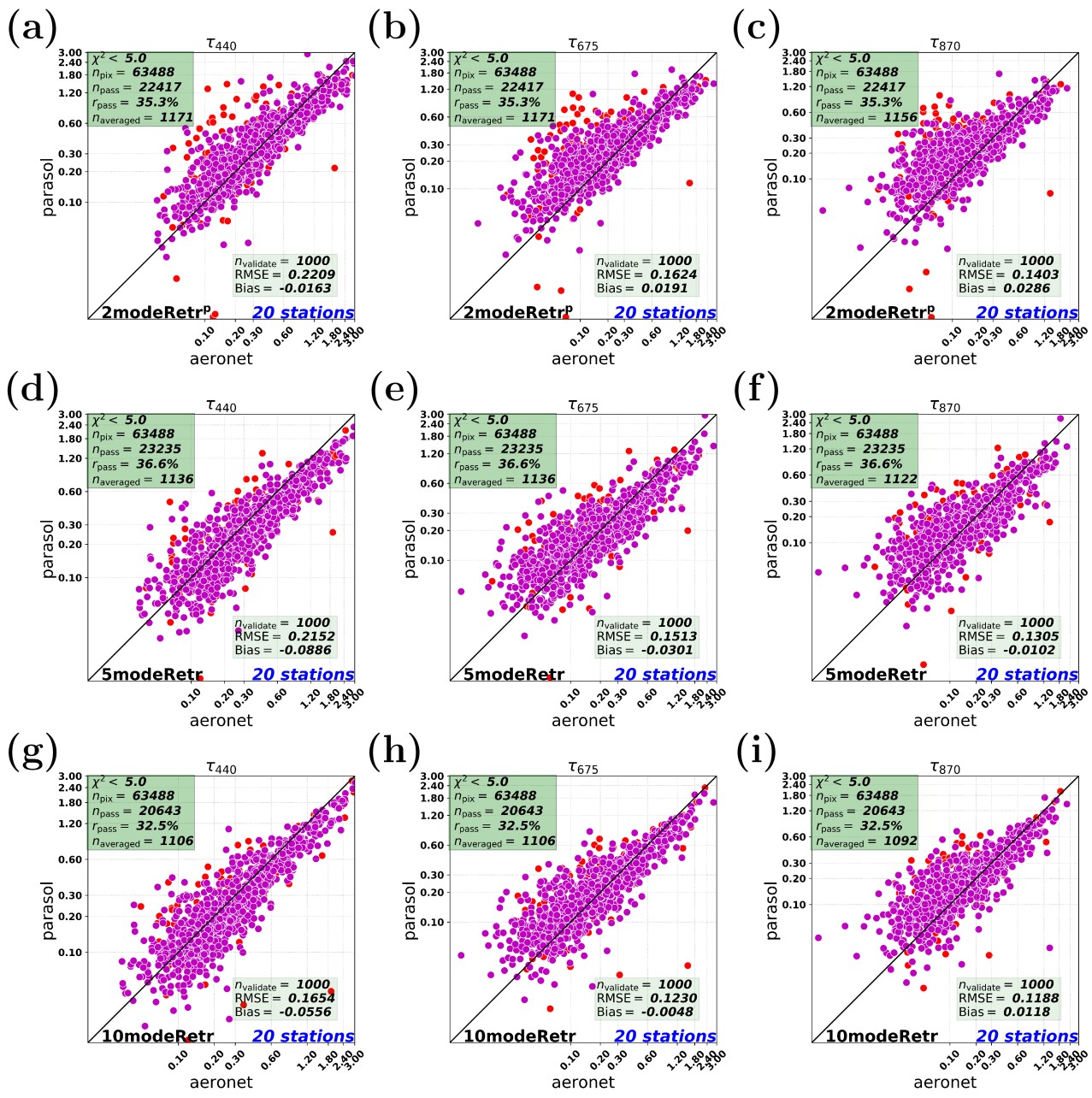

**Figure 13. Real data retrievals of AOT among 2modeRetr[p], 5modeRetr and 10modeRetr at different wavelengths.** The red and magenta points represent $n_{averaged}$ and $n_{validate}$ points, respectively. **(a)**, **(b)**, **(c)** are 2modeRetr[p] at 440 nm, 675 nm, 870 nm, respectively. **(d)**, **(e)**, **(f)** are 5modeRetr at 440 nm, 675 nm, 870 nm, respectively. **(g)**, **(h)**, **(i)** are 10modeRetr at 440 nm, 675 nm, 870 nm, respectively.

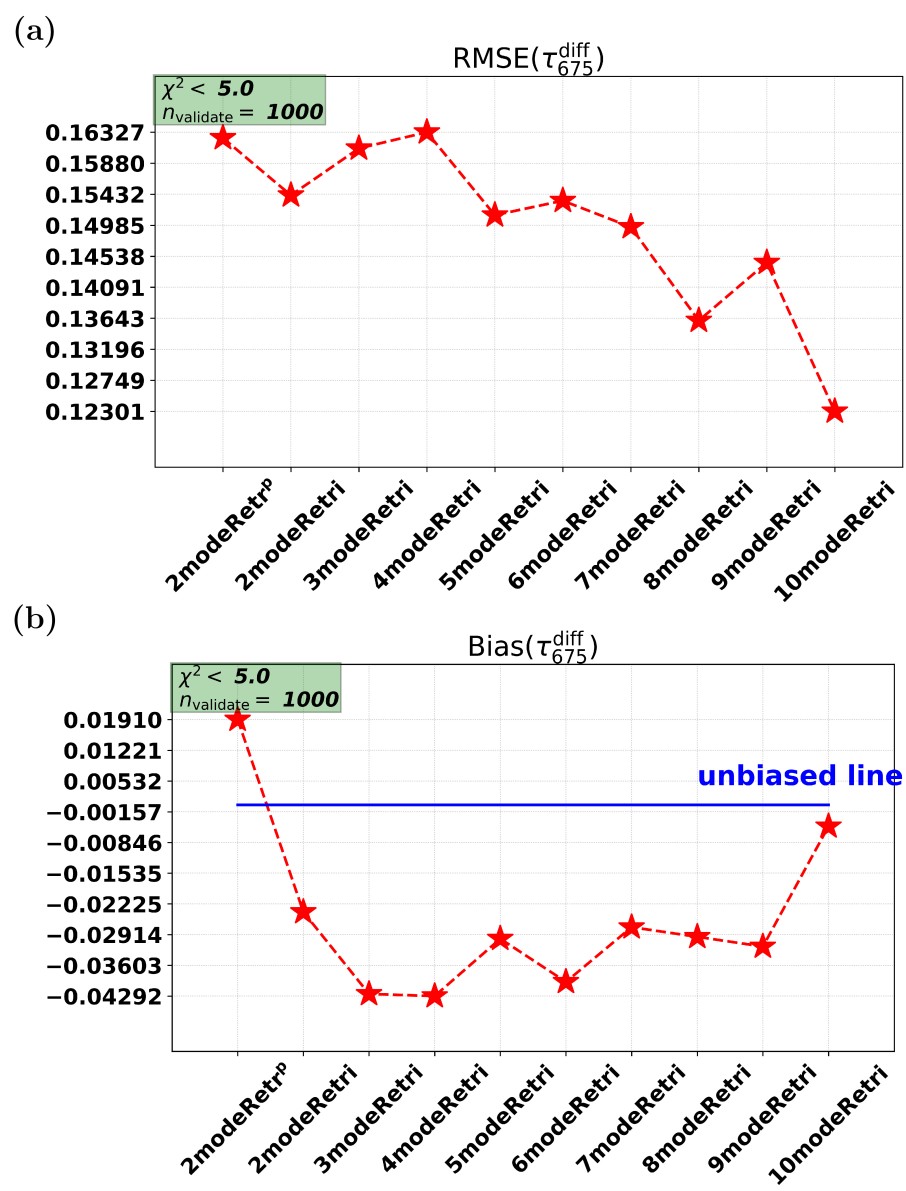

**Figure 14. Real data retrievals for AOT: Root-Mean-Square error (RMSE) and bias for the difference between PARASOL retrievals and AERONET data.**

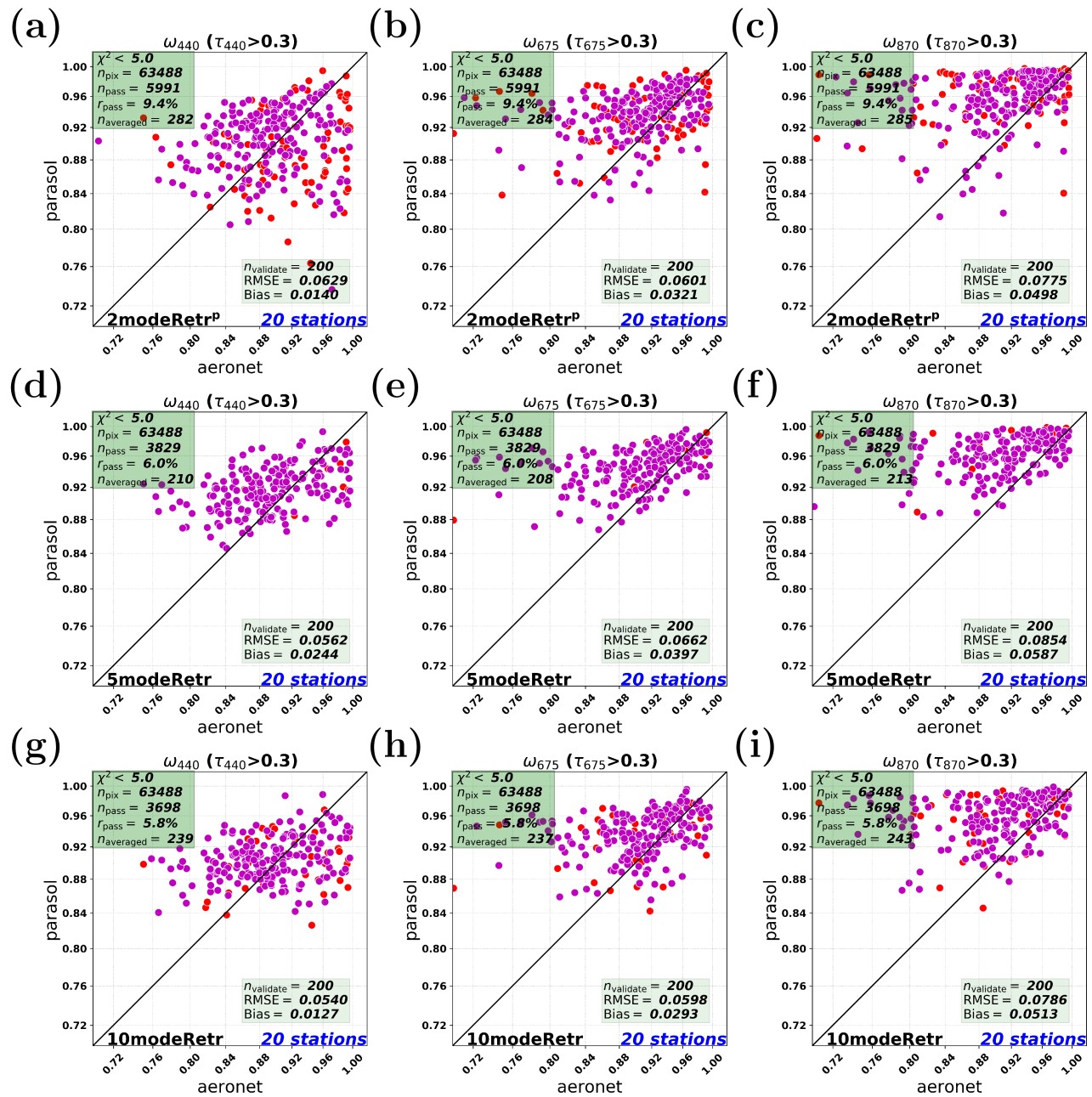

**Figure 15. Real data retrievals of SSA among 2modeRetr$^{\text{P}}$, 5modeRetr and 10modeRetr at different wavelengths.** The red and magenta points represent $n_{\text{averaged}}$ and $n_{\text{validate}}$ points, respectively. **(a)**, **(b)**, **(c)** are 2modeRetr$^{\text{P}}$ at 440 nm, 675 nm, 870 nm, respectively. **(d)**, **(e)**, **(f)** are 5modeRetr at 440 nm, 675 nm, 870 nm, respectively. **(g)**, **(h)**, **(i)** are 10modeRetr at 440 nm, 675 nm, 870 nm, respectively.

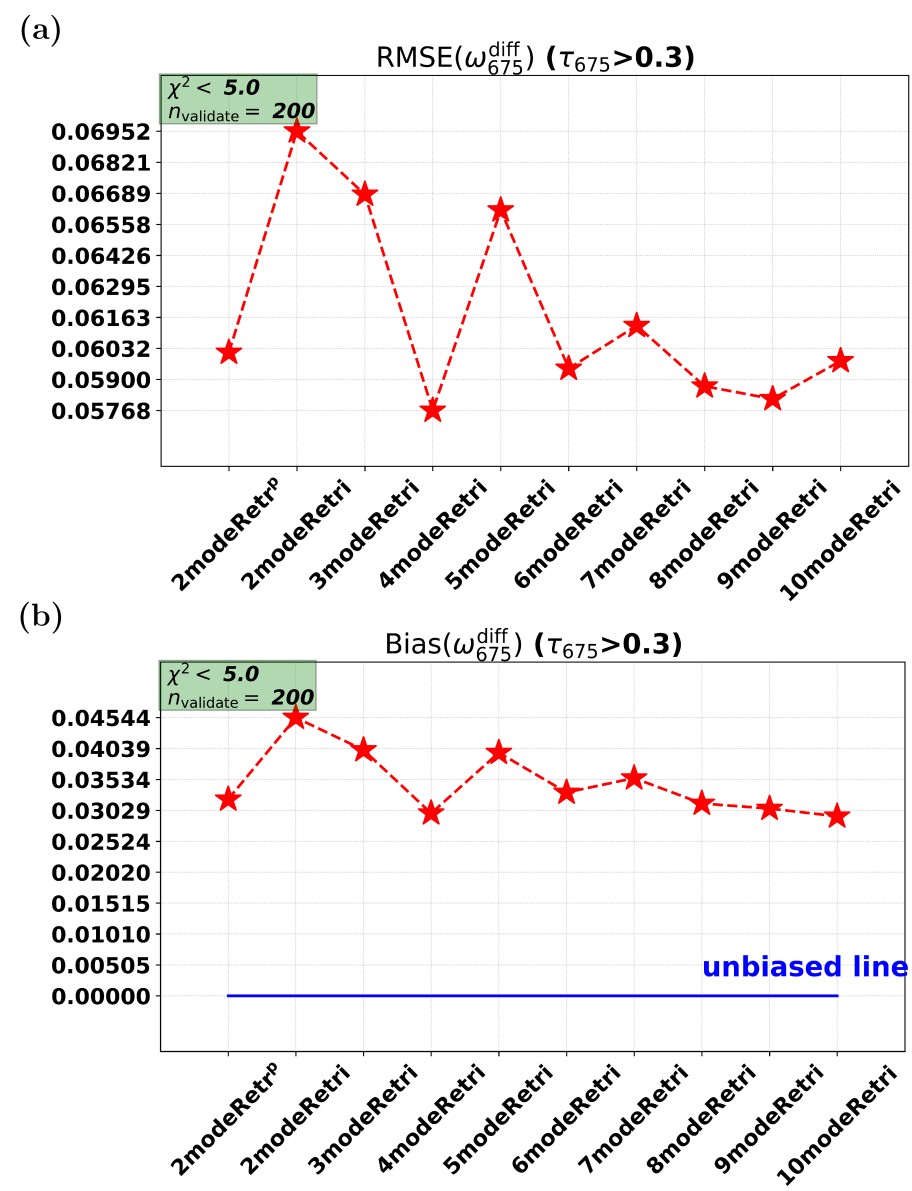

**Figure 16. Real data retrievals for SSA: Root-Mean-Square error (RMSE) and bias for the difference between PARASOL retrievals and AERONET data.**

**Table 1.** Accuracy Requirements on aerosol properties from Mishchenko et al. (2004) as used for the Glory mission, Global Climate Observing System (GCOS), and the ACE study (https://acemission.gsfc.nasa.gov/documents/ACE_Report5_Aerosol_Science_v7.pdf).

| Property | Glory | GCOS | ACE |
|---|---|---|---|
| AOT | max(0.04, 10 %) | max(0.03, 10 %) | max(0.02, 5 %) |
| SSA | 0.03 | 0.03 | 0.02 |
| $r_{\mathrm{eff}}$ | max(0.1 $\mu m$, 10 %) | - | 10 % |
| $v_{\mathrm{eff}}$ | max(0.3, 50 %) | - | 50 % |
| $m_{\mathrm{r}}$ | 0.02 | - | 0.02 |
| $N$ | - | - | 100 % |
| ALH | - | 1000 m | 500 m |

**Table 2.** Multi-mode retrieval definition.

| | mode 1 | mode 2 | mode 3 | mode 4 | mode 5 | mode 6 | mode 7 | mode 8 | mode 9 | mode 10 |
|---|---|---|---|---|---|---|---|---|---|---|
| $r_{\mathrm{eff}}$ ($\mu m$) | 0.070 | 0.094 | 0.130 | 0.163 | 0.220 | 0.282 | 0.882 | 1.2 | 1.759 | 3.0 |
| $v_{\mathrm{eff}}$ | 0.130 | 0.130 | 0.130 | 0.130 | 0.130 | 0.130 | 0.284 | 1.0 | 1.718 | 1.718 |
| $f_{\mathrm{sphere}}$ | 1.0 | 1.0 | 1.0 | 1.0 | 1.0 | 1.0 | | | | |
| **10-mode retrieval** | × | × | × | × | × | × | × | × | × | × |
| **9-mode retrieval** | × | × | × | × | × | × | × | | × | × |
| **8-mode retrieval** | × | × | × | × | | × | × | | × | × |
| **7-mode retrieval** | × | × | | × | | × | × | | × | × |
| **6-mode retrieval** | × | × | | × | | × | × | | × | |
| **5-mode retrieval** | | × | | × | | × | × | | × | |
| **4-mode retrieval** | | × | | × | | × | | | × | |
| **3-mode retrieval** | | × | | × | | | | | × | |
| **2-mode retrieval** | | | | × | | | | | × | |

**Table 3.** Abbreviations for different retrieval cases

| Retrieval cases | Abbreviation |
|---|---|
| Parametric 2-mode retrieval | 2modeRetr$^{\mathrm{P}}$ |
| Multi(2)-mode retrieval | 2modeRetr |
| Multi(3)-mode retrieval | 3modeRetr |
| Multi(4)-mode retrieval | 4modeRetr |
| Multi(5)-mode retrieval | 5modeRetr |
| Multi(6)-mode retrieval | 6modeRetr |
| Multi(7)-mode retrieval | 7modeRetr |
| Multi(8)-mode retrieval | 8modeRetr |
| Multi(9)-mode retrieval | 9modeRetr |
| Multi(10)-mode retrieval | 10modeRetr |

**Table 4.** State vector for parametric 2-mode retrieval and multi-mode retrieval.

| | Parameters in the state vector | Parametric 2-mode retrieval | Multi-mode retrieval |
|---|---|---|---|
| | Effective radius | $r_{\mathrm{eff}}^{\mathrm{f}}, r_{\mathrm{eff}}^{\mathrm{c}}$ | \ |
| | Effective variance | $v_{\mathrm{eff}}^{\mathrm{f}}, v_{\mathrm{eff}}^{\mathrm{c}}$ | \ |
| Aerosol | Aerosol loading | $N^{\mathrm{f}}, N^{\mathrm{c}}$ | $N^j, (j=1,2,...,n_{\mathrm{mode}})$ |
| properties | fraction of spheres | $f_{\mathrm{sphere}}^{\mathrm{c}}$ | $f_{\mathrm{sphere}}^{\mathrm{c}}$ |
| | Mode component coefficients | $\alpha_k^{\mathrm{f}}, \alpha_k^{\mathrm{c}}, (k=1,2)$ | $\alpha_k^{\mathrm{f}}, \alpha_k^{\mathrm{c}}, (k=1,2)$ |
| | Aerosol layer height | $z$ | $z$ |
| | Scaling parameter for BPDF model | $x_{\mathrm{bpdf}}^{\mathrm{scale}}$ | $x_{\mathrm{bpdf}}^{\mathrm{scale}}$ |
| Surface | Coefficient of LI Sparse kernel | $x_{\mathrm{bdrf}}^{\mathrm{geo1}}$ | $x_{\mathrm{bdrf}}^{\mathrm{geo1}}$ |
| properties | Coefficient of ROSS Thick kernel | $x_{\mathrm{bdrf}}^{\mathrm{geo2}}$ | $x_{\mathrm{bdrf}}^{\mathrm{geo2}}$ |
| | BDRF scaling parameters for wavelength bands | $x_{\mathrm{bdrf}}^{iw}, (iw=1,2,\cdots,n_{\mathrm{wave}})$ | $x_{\mathrm{bdrf}}^{iw}, (iw=1,2,\cdots,n_{\mathrm{wave}})$ |
| | Number of aerosol parameters | 12 | $n_{\mathrm{mode}}$+6 |
| | Number of surface parameters | $n_{\mathrm{wave}}$+3 | $n_{\mathrm{wave}}$+3 |
| | Length of the state vector | $n_{\mathrm{wave}}$+15 | $n_{\mathrm{mode}}$+$n_{\mathrm{wave}}$+9 |

**Table 5.** Prior values and weighting factors for the state vector in the parametric 2-mode retrieval and the multi-mode retrieval. The prior value and weighting factor of aerosol loading $N$ for each mode are further calculated based on Mie theory using the prior information of $\tau^{\mathrm{f}}$ and $\tau^{\mathrm{c}}$ from the table.

| Elements | Prior values | | Weighting factors | |
|---|---|---|---|---|
| | Parametric 2-mode | Multi-mode | Parametric 2-mode | Multi-mode |
| $r_{\mathrm{eff}}^{\mathrm{f}}$ ($\mu$m) | 0.2 | \ | $0.1^2$ | \ |
| $r_{\mathrm{eff}}^{\mathrm{c}}$ ($\mu$m) | 1.5 | \ | $1.0^2$ | \ |
| $v_{\mathrm{eff}}^{\mathrm{f}}$ | 0.2 | \ | $0.05^2$ | \ |
| $v_{\mathrm{eff}}^{\mathrm{c}}$ | 0.6 | \ | $0.1^2$ | \ |
| $\tau^{\mathrm{f}}$ | 0.2 | 0.001 | $2.0^2$ | $\left(\frac{0.5}{n_{\mathrm{mode}}}\right)^2$ |
| $\tau^{\mathrm{c}}$ | 0.05 | 0.001 | $2.0^2$ | $\left(\frac{1.0}{n_{\mathrm{mode}}}\right)^2$ |
| $\alpha_1^{\mathrm{f}}$ | 0.9 | | $0.1^2$ | |
| $\alpha_2^{\mathrm{f}}$ | 0.005 | | $0.1^2$ | |
| $\alpha_1^{\mathrm{c}}$ | 0.5 | | $0.1^2$ | |
| $\alpha_2^{\mathrm{c}}$ | 0.5 | | $0.1^2$ | |
| $f_{\mathrm{sphere}}^{\mathrm{c}}$ | 0.95 | | $1.0^2$ | |
| $z$ (km) | 2.0 | | $4.0^2$ | |
| $x_{\mathrm{bpdf}}^{\mathrm{scale}}$ | 4.0 | | $5.0^2$ | |
| $x_{\mathrm{bdrf}}^{\mathrm{geo1}}$ (LI kernel) | 0.0 | | $0.25^2$ | |
| $x_{\mathrm{bdrf}}^{\mathrm{geo2}}$ (ROSS kernel) | 0.0 | | $1.0^2$ | |
| $x_{\mathrm{bdrf}}^{iw}$, ($iw = 1, 2, \cdots, n_{\mathrm{wave}}$) | 0.0 | | $0.5^2$ | |

**Table 6.** Parameters to create a 10-mode lookup table.

| Parameters | mode 1 | mode 2 | mode 3 | mode 4 | mode 5 | mode 6 | mode 7 | mode 8 | mode 9 | mode 10 |
|---|---|---|---|---|---|---|---|---|---|---|
| $r_{\mathrm{eff}}$ ($\mu m$) | 0.070 | 0.094 | 0.130 | 0.163 | 0.220 | 0.282 | 0.882 | 1.2 | 1.759 | 3.0 |
| $v_{\mathrm{eff}}$ | 0.130 | 0.130 | 0.130 | 0.130 | 0.130 | 0.130 | 0.284 | 1.0 | 1.718 | 1.718 |
| $f_{\mathrm{sphere}}$ | | | 1.0 | | | | | 0.5 | | |
| $m_{\mathrm{r}}$ (550 $nm$) | | | | | 1.45 | | | | | |
| $m_{\mathrm{i}}$ (550 $nm$) | | | | | 0.02 | | | | | |
| $\tau$ | | | | 0.01, 0.15, 0.25, 0.5, 0.8, 1.0, 1.5, 3.0, 5.0 | | | | | | |
| $z$ (km) | | | | | 2.0 | | | | | |

| | |
|---|---|
| Wavelength bands (nm) | 390.0, 400.0, 410.0, 440.0, 450.0, 470.0, 491.5, 500.0, 550.0, 565.0, 600.0, 670.0, 750.0, 863.4, 1019.4 |
| VZA (degree) | 0, 10.0, 20.0, 30.0, 40.0, 50.0, 65.0 |
| SZA (degree) | 10.0, 15.0, 20.0, 25.0, 30.0, 35.0, 40.0, 45.0, 50.0, 55.0, 60.0, 65.0, 70.0, 75.0 |
| Surface pressure (mbar) | 700.0, 1013.0 |
| $x_{\mathrm{bpdf}}^{\mathrm{scale}}$ | 1.0, 8.0 |
| $x_{\mathrm{bdrf}}^{\mathrm{geo1}}$ (LI kernel) | 0, 0.1, 0.2 |
| $x_{\mathrm{bdrf}}^{\mathrm{geo2}}$ (ROSS kernel) | 0, 0.5, 1.0, 1.5 |
| $x_{\mathrm{bdrf}}^{iw}$, $(iw = 1, 2, \cdots, 11)$ | 0.01, 0.015, 0.02, 0.03, 0.05, 0.07, 0.1 |
| $x_{\mathrm{bdrf}}^{iw}$, $(iw = 12)$ | 0.01, 0.015, 0.025, 0.06, 0.075, 0.1, 0.125 |
| $x_{\mathrm{bdrf}}^{iw}$, $(iw = 13)$ | 0.05, 0.07, 0.1, 0.13, 0.175, 0.25, 0.35 |
| $x_{\mathrm{bdrf}}^{iw}$, $(iw = 14, 15)$ | 0.1, 0.15, 0.2, 0.25, 0.35, 0.4, 0.5 |

**Table 7.** AERONET stations for validation of PARASOL retrievals

| | | | | |
|---|---|---|---|---|
| Alta_Floresta | Ames | BONDVILLE | Bac_Giang | Banizoumbou |
| Belsk | Cabauw | Chiang_Mai_Met_Sta | Fontainebleau | Fresno |
| Kanpur | Lille | Minsk | Mongu | Moscow_MSU_MO |
| Mukdahan | Trinidad_Head | Zvenigorod | XiangHe | Beijing |