# Peer review of "Retrieval of aerosol micro-physical and optical properties over land using a multi-mode approach"

_Atmospheric Measurement Techniques, 2018_

## Referee Comment (RC1) · M. Mishchenko (Referee) · 20 Oct 2018

While this paper is neither exhaustive nor definitive, it can safely be called very useful as it adds valuable insights that should help instrument teams adopt alternative and potentially more appropriate retrieval strategies. I recommend publication after the few (mostly minor) comments listed below have been addressed.

1. The authors should make it maximally clear that the results of their study are likely to be heavily preconditioned by their choice of POLDER-like measurements (real or synthetic). The measurement accuracies selected for both intensity and polarization are hardly realistic. There is no discussion of what could happen if, e.g., an APS-type dataset were used, with its higher accuracy, more scattering angles, and wider spectral

[Figure]

range (e.g., Mishchenko, M. I., B. Cairns, G. Kopp, C. F. Schueler, B. A. Fafaul, J. E. Hansen, R. J. Hooker, T. Itchkawich, H. B. Maring, and L. D. Travis, 2007: Accurate monitoring of terrestrial aerosols and total solar irradiance: introducing the Glory Mission. Bull. Amer. Meteorol. Soc. 88, 677–691). Furthermore, the paper is focused on what and how well can be retrieved without putting the outcome in the context of what and how accurately must in fact be retrieved (e.g., Mishchenko, M. I., B. Cairns, J. E. Hansen, L. D. Travis, R. Burg, Y. J. Kaufman, J. V. Martins, and E. P. Shettle, 2004: Monitoring of aerosol forcing of climate from space: analysis of measurement requirements. J. Quant. Spectrosc. Radiat. Transfer 88, 149–161). At least a short discussion of these important aspects is desirable.

2. As far as I understand, the BC aerosols are in the accumulation category and hence are treated as homogeneous spheres. If that's the case, the authors should at least acknowledge that this treatment can be exceedingly unrealistic (e.g., Liu, L., and M. I. Mishchenko, 2018: Scattering and radiative properties of morphologically complex carbonaceous aerosols: a systematic modeling study. Remote Sens. 10, 1634).

3. "All fine modes are assumed to have the same refractive index and all coarse modes have another refractive index value". This assumption is highly artificial. Can its robustness be somehow checked by mixing BC and sulfate aerosols with their actual refractive indices and then performing a synthetic retrieval assuming that the refractive indices are the same? What would be the meaning and usefulness of such a retrieval?

4. The sentence "We use the Mie/T-Matrix approach of Dubovik et al. (2006) with their proposed spheroid aspect ratio distribution for computing optical properties for a mixture of spheroids and spheres" can be made factually more accurate. For example, "Nonspherical aerosols are modeled as a size/shape mixture of randomly oriented spheroids (Hill, S. C., Hill, A. C., and Barber, P. W., 1984: Light scattering by size/shape distributions of soil particles and spheroids. Appl. Opt. 23, 1025–1031. Mishchenko, M. I., L. D. Travis, R. A. Kahn, and R. A. West, 1997: Modeling phase functions for dustlike tropospheric aerosols using a shape mixture of randomly oriented polydisperse spheroids. J. Geophys. Res. 102, 16831–16847). We use the Mie/T-Matrix/Improved-Geometrical-Optics database by Dubovik et al. (2006) along with their proposed spheroid aspect ratio distribution for computing optical properties for a mixture of spheroids and spheres."

5. "Number column" should be "column number" throughout.
* * *

---

## Referee Comment (RC2) · R. Lang (Referee) · 2 Nov 2018

The paper by Fu and Hasekamp provides very useful insight in the different performance of retrievals applying a parameterization of the 2-mode (fine and coarse) size distribution on the one hand (parametric mode), and retrievals utilizing more than two modes of size distribution parameters (multi-mode retrievals). I can recommend the paper for publication because it addresses a fundamental choice made by a lot of aerosol retrievals, which may have significant consequences not only for the accuracy of the retrievals (as the authors show) but potentially also for its computational cost. The paper may therefore provide some guidance for selecting an appropriate retrieval method in the context of accuracy needs, computational performance and the available information content in the measurement. Fu and Hasekamp are basing their results of the

evaluation of the performance of the both mode-fitting approaches on the same SRON based RT and inversion algorithm using both synthetic and PARASOL measurement. The latter are then evaluated with respect to Aeronet data.

While the presented approach is solid in terms of performance differences for the SRON Algorithm, what is missing is a discussion to which extend the results are significant even for other full inversion approaches and RT models. Since there are currently not many which can perform at a global scale (as the authors state themselves) it would have been interesting to understand if these few algorithms would converge in their performance when using the same type of mode fitting. While I would guess that the authors ultimately had such a comparison in mind, a discussion of the results in this context - and maybe providing some outlook on how to apply this kind of sensitivity test also in the context of other retrieval schemes - is currently missing.

Before publication I would like to ask the authors to also address the following set of comments:

1) While one would expect that retrievals over water surfaces would reduce the parameter space and therefore may make the evaluation of the performance difference for different mode fitting scheme more robust, the study focuses on land surfaces only without further qualification. In this context I am also missing a discussion of the combination of wavelength, surface properties and scattering geometry, on the synthetic results, since some of the combinations may not be realistic and may therefore complicate the interpretation of the comparison of the synthetic results to the performance using PARASOL measurements. Ideally the synthetic retrievals could limited to the observation geometries and surface properties combination available at the Aeronet stations. This greatly would improve the interpretation of the PARASOL retrieval results in the context of the synthetic retrievals.

2) The difference of the retrieval performance between consistent and inconsistent synthetic retrievals are potentially very interesting to understand the frequent problems

when using, for example, generic pre-launch TOA test-data sets for end-to-end system performance studies and developments. However a more detailed interpretation or analysis of the results appears to be missing in the paper. The results for AOT at least seem to indicate that parametric 2 mode retrievals perform better in inconsistent cases than multi-mode retrievals. Can this be understood or explained?

3) For the PARASOL retrievals in section 5 it is stated that multi-mode retrials with more than 4 modes perform well (while at the same time mode-5 seems to have the largest bias), whereas the conclusion from the synthetic retrievals was that multi-mode retrievals perform well for n>5. Could there be a reason for this (although small) discrepancy. Overall the results are presented as if mode-5 is a kind of physical significant lower limit for multi-mode retrievals (if yes, why?), while the results seem to more indicate a general trend for decreasing RMS with higher mode numbers.

4) In Section 3.1 there is a reference missing to the actual PARASOL data and its version used. Are there references available for the expected intensity and polarisation error for PARASOL?

---

## Author Response (AR1)

Dear Editor,

Herewith we submit the revised manuscript. First we would like to thank you and Dr. Mishchenko and Dr. Lang, and appreciate all the comments and suggestions. We have considered all of them and made changes accordingly in the revised paper.

In the following we will give our responses to the comments. To make the changes easier to identify, we have marked them with different colors.

Kind regards,
Guangliang Fu and Otto Hasekamp

(The revised manuscript is in the latter part of this pdf.)

**Responses – part 1:**

**Reply to comments**:

1. *The authors should make it maximally clear that the results of their study are likely to be heavily preconditioned by their choice of POLDER-like measurements (real or synthetic). The measurement accuracies selected for both intensity and polarization are hardly realistic. There is no discussion of what could happen if, e.g., an APS-type dataset were used, with its higher accuracy, more scattering angles, and wider spectral range (e.g., Mishchenko, M. I., B. Cairns, G. Kopp, C. F. Schueler, B. A. Fafaul, J. E. Hansen, R. J. Hooker, T. Itchkawich, H. B. Maring, and L. D. Travis, 2007: Accurate monitoring of terrestrial aerosols and total solar irradiance: introducing the Glory Mission. Bull. Amer. Meteorol. Soc. 88, 677–691). Furthermore, the paper is focused on what and how well can be retrieved without putting the outcome in the context of what and how accurately must in fact be retrieved (e.g., Mishchenko, M. I., B. Cairns, J. E. Hansen, L. D. Travis, R. Burg, Y. J. Kaufman, J. V. Martins, and E. P. Shettle, 2004: Monitoring of aerosol forcing of climate from space: analysis of measurement requirements. J. Quant. Spectrosc. Radiat. Transfer 88, 149–161). At least a short discussion of these important aspects is desirable.*

   Response:
   We agree that the results are (partly) pre-conditioned by the choice of POLDER, especially for real measurements. We added a phrase to the paper in Sect. 3.1:
   " It should also be noted that higher accuracy aerosol retrievals are to be expected for all parameters from instruments that have higher polarimetric accuracy, more scattering angles and/or spectral bands (e.g. (Mishchenko and Travis, 1997; Hasekamp and Landgraf, 2007)). Examples of such improved instruments are GLORY-APS (Mishchenko et al., 2007), MAIA (Diner et al., 2018), SPEXone (Hasekamp et al., 2018), and HARP-2 (Martins et al., 2017). "

   We expect that the synthetic results are less affected by the choice for the POLDER setup because the synthetic retrievals have been performed on noise-free data. Given that the 'consistent' retrievals already look close to perfect for these synthetic measurement, there is hardly room for improvement by bringing in extra measurements.

   Concerning the comment about requirements, we now include a Table in the paper that lists requirements from different sources (APS-GLORY, GCOS, ACE). Now the readers can see the results in perspective of these requirements.

Table 1: Accuracy Requirements on aerosol properties from Mishchenko et al. (2004) as used for the Glory mission, Global Climate Observing System (GCOS), and the ACE study (`https://acemission.gsfc.nasa.gov/documents/ACE_Report5_Aerosol_Science_v7.pdf`).

| Property | Glory | GCOS | ACE |
|---|---|---|---|
| AOT | max(0.04, 10 %) | max(0.03, 10 %) | max(0.02, 5 %) |
| SSA | 0.03 | 0.03 | 0.02 |
| $r_{\text{eff}}$ | max(0.1 $\mu m$, 10 %) | - | 10 % |
| $v_{\text{eff}}$ | max(0.3, 50 %) | - | 50 % |
| $m_{\text{r}}$ | 0.02 | - | 0.02 |
| $N$ | - | - | 100 % |
| ALH | - | 1000 m | 500 m |

2. *As far as I understand, the BC aerosols are in the accumulation category and hence are treated as homogeneous spheres. If that's the case, the authors should at least acknowledge that this treatment can be exceedingly unrealistic (e.g., Liu, L., and M. I. Mishchenko, 2018: Scattering and radiative properties of morphologically complex carbonaceous aerosols: a systematic modeling study. Remote Sens. 10, 1634).*

Response:
Yes. We have added a phrase in Sect. 2.2:
" (A recent study by Liu and Mishchenko (2018) indicates that this assumption becomes unrealistic for increasing fraction of carbonaceous aerosol in the fine mode.) "

3. *"All fine modes are assumed to have the same refractive index and all coarse modes have another refractive index value". This assumption is highly artificial. Can its robustness be somehow checked by mixing BC and sulfate aerosols with their actual refractive indices and then performing a synthetic retrieval assuming that the refractive indices are the same? What would be the meaning and usefulness of such a retrieval?*

Response:
Although probably not fully realistic, the separation into a 'fine mode refractive index' and 'coarse mode refractive index' is quite common in polarimetric aerosol retrievals (e.g. (Chowdhary et al., 2001; Waquet et al., 2009; Hasekamp et al., 2011)). Also aerosol retrievals have been performed using an even simpler assumption of a constant refractive index for all modes (e.g. (Dubovik et al., 2011; Xu et al., 2017)).

It seems that the sentence did not clearly explain the approach. Therefore, we changed " all fine modes are assumed to have the same refractive index and all coarse modes have another refractive index value. "
to (in Sect. 4.5.1):

" also for multi-mode retrievals we use a separate refractive index for the fine and coarse mode, respectively. In this case, the fine mode refractive index corresponds to mode number 1-6 in Table 2 and the coarse mode refractive index to mode 7-10. "

4. *The sentence "We use the Mie/T-Matrix approach of Dubovik et al. (2006) with their proposed spheroid aspect ratio distribution for computing optical properties for a mixture of spheroids and spheres" can be made factually more accurate. For example, "Nonspherical aerosols are modeled as a size/shape mixture of randomly oriented spheroids (Hill, S. C., Hill, A. C., and Barber, P. W., 1984: Light scattering by size/shape distributions of soil particles and spheroids. Appl. Opt. 23, 1025–1031. Mishchenko, M. I., L. D. Travis, R. A. Kahn, and R. A. West, 1997: Modeling phase functions for dustlike tropospheric aerosols using a shape mixture of randomly oriented polydisperse spheroids. J. Geophys. Res. 102, 16831–16847). We use the Mie/T-Matrix/Improved-Geometrical-Optics database by Dubovik et al. (2006) along with their proposed spheroid aspect ratio distribution for computing optical properties for a mixture of spheroids and spheres."*

Response:

We changed the description to (in Sect. 2.1):

" Nonspherical aerosols are modeled as a size/shape mixture of randomly oriented spheroids (Hill et al., 1984; Mishchenko et al., 1997). We use the Mie/T-Matrix/Improved-Geometrical-Optics database by Dubovik et al. (2006) along with their proposed spheroid aspect ratio distribution for computing optical properties for a mixture of spheroids and spheres. "

5. *"Number column" should be "column number" throughout.*

Response:

It has been corrected throughout the manuscript.

**Responses – part 2:**

**Reply to General comments**:

1. *The paper by Fu and Hasekamp provides very useful insight in the different performance of retrievals applying a parameterization of the 2-mode (fine and coarse) size distribution on the one hand (parametric mode), and retrievals utilizing more than two modes of size distribution parameters (multi-mode retrievals). I can recommend the paper for publication because it addresses a fundamental choice made by a lot of aerosol retrievals, which may have significant consequences not only for the accuracy of the retrievals (as the authors show) but potentially also for its computational cost. The paper may therefore provide some guidance for selecting an appropriate retrieval method in the context of accuracy needs, computational performance and the available information content in the measurement. Fu and Hasekamp are basing their results of the evaluation of the performance of the both mode-fitting approaches on the same SRON based RT and inversion algorithm using both synthetic and PARASOL measurement.*

   *The latter are then evaluated with respect to Aeronet data. While the presented approach is solid in terms of performance differences for the SRON Algorithm, what is missing is a discussion to which extend the results are significant even for other full inversion approaches and RT models. Since there are currently not many which can perform at a global scale (as the authors state themselves) it would have been interesting to understand if these few algorithms would converge in their performance when using the same type of mode fitting. While I would guess that the authors ultimately had such a comparison in mind, a discussion of the results in this context - and maybe providing some outlook on how to apply this kind of sensitivity test also in the context of other retrieval schemes - is currently missing.*

   Response:
   The performance of a retrieval algorithm depends on a number of things in addition to the state vector definition for aerosols (studied here): The inversion approach (cost function, regularization strength, multi- versus single pixel), the accuracy of the forward model, the surface reflection model. Now the SRON algorithm is extended to an arbitrary number of 'fixed' modes, it can be more easily compared to e.g. the GRASP algorithm which is also based on fixed modes. For such comparisons, it is important to study, where possible, the algorithm differences in a systematic manner. This would include using the same surface reflection model and ensuring that the RT models in both algorithms agree with benchmark

results.

We added the following paragraph to the 'Discussion and Conclusion' section:
" When comparing retrievals between different algorithms, it is important to realize that the performance of a given algorithm depends on a number of factors, the definition of the aerosol state vector being one of them. Other factors are the inversion approach (cost function, regularization strength, multiple versus single pixel), the accuracy of the forward model, and the surface reflection model. It is important to study the above mentioned aspects with an individual algorithm. However, now that the SRON algorithm has been extended to include an arbitrary number of fixed modes, it has become easier to compare to other algorithms using a similar state vector definition (Dubovik et al., 2011; Xu et al., 2017). This would be an important topic for future research. "

**Reply to specific comments**:

1. *While one would expect that retrievals over water surfaces would reduce the parameter space and therefore may make the evaluation of the performance difference for different mode fitting scheme more robust, the study focuses on land surfaces only without further qualification. In this context I am also missing a discussion of the combination of wavelength, surface properties and scattering geometry, on the synthetic results, since some of the combinations may not be realistic and may therefore complicate the interpretation of the comparison of the synthetic results to the performance using PARASOL measurements. Ideally the synthetic retrievals could limited to the observation geometries and surface properties combination available at the Aeronet stations. This greatly would improve the interpretation of the PARASOL retrieval results in the context of the synthetic retrievals.*

   Response:
   – To make explicitly clear that the paper relates to aerosol retrievals over land, we changed the title to:
   " Retrieval of aerosol micro-physical and optical properties over land using a multi-mode approach ".

   –We believe that the surface properties in our synthetic measurements are representative for those over AERONET sites. However, as noted by the reviewer, the viewing geometries by which a certain ground pixel is seen by POLDER-3 varies drastically over season and over the orbit/swath. The geometry used in the paper is one of the 'best' geometries as it assumes measurements in the principal plane. To investigate the effect of this assumption we also performed the same study for a more 'difficult' geometry with high sun and a relative azimuth angle of 60/−120 degrees, i.e. far away from the principal plane. The

geometry used in the paper (GEOM-1) and the more 'difficult' one (GEOM-2) are summarized in Table 1 of this response. Figures 17– 23 show the result of the synthetic study for GEOM-2 for AOT, SSA, real refractive index (fine and coarse), imaginary refractive index (fine and coarse), and aerosol layer height. These results can be compared to the corresponding results for GEOM-1 in the paper. Although the overall errors are larger for GEOM-2 on most parameters, the general conclusions of the synthetic study still hold for GEOM-2. Therefore, we are convinced that they are representative. We added the following phrase to the paper in Section 'Discussions and conclusions':

" It should be noted that the geometry used for the synthetic study in this paper is quite favorable as it assumes measurements in the principal plane. We also did the same synthetic study for a much less favorable geometry (SZA=20°, relative azimuth angle=60°/-120°). Although for the latter geometry, the performance is somewhat worse, the main conclusions from the synthetic study still hold for this geometry. "

Table 1: Different observation geometries in synthetic retrievals. (GEOM-1 corresponds to synthetic retrievals in the paper; GEOM-2 is for a test of effect of different geometry.)

| | GEOM-1 | GEOM-2 |
|---|---|---|
| Solar Zenith Angle (SZA) (degree) | 41.0 | 20.0 |
| Satellite Viewing Zenith Angle (VZA) (degree) | 60.0, 50.0, 40.0, 30.0, 20.0, 10.0, 0.0, -10.0, -20.0, -30.0, -40.0, -50.0, -60.0 | |
| Satellite Azimuth Angle (SAA) (degree) | 180.0 (if VZA<0) 0.0 (if VZA>0) | 120.0 (if VZA<0) 60.0 (if VZA>0) |
| Scatter Angle (ScatA) (degree) | 79.0, 89.0, 99.0, 109.0, 119.0, 129.0, 139.0, 149.0, 159.0, 169.0, 179.0, 171.0, 161.0 | 108.8, 118.2, 127.6, 136.7, 145.5, 153.6, 160.0, 162.8, 160.3, 154.1, 146.1, 137.3, 128.2 |

[Figure]

Figure 17: **Synthetic retrievals using GEOM-2 for AOT.**

[Figure]

Figure 18: **Synthetic retrievals using GEOM-2 for SSA.**

[Figure]

Figure 19: **Synthetic retrievals using GEOM-2 for the real part of refractive index (at 550 nm) of the fine modes ($m_{\mathrm{r}}^{\mathrm{f}}$).**

[Figure]

Figure 20: **Synthetic retrievals using GEOM-2 for the real part of refractive index (at 550 nm) of the coarse modes ($m_{\mathrm{r}}^{\mathrm{c}}$).**

[Figure]

Figure 21: **Synthetic retrievals using GEOM-2 for the imaginary part of refractive index (at 550 nm) of the fine modes ($m_i^f$).**

[Figure]

Figure 22: **Synthetic retrievals using GEOM-2 for the imaginary part of refractive index (at 550 nm) of the coarse modes ($m_i^c$).**

[Figure]

Figure 23: **Synthetic retrievals using GEOM-2 for the central height ($z$) of the aerosol layer.**

2. *The difference of the retrieval performance between consistent and inconsistent synthetic retrievals are potentially very interesting to understand the frequent problems when using, for example, generic pre-launch TOA test-data sets for end-to-end system performance studies and developments. However a more detailed interpretation or analysis of the results appears to be missing in the paper. The results for AOT at least seem to indicate that parametric 2 mode retrievals perform better in inconsistent cases than multi-mode retrievals. Can this be understood or explained?*

Response:

Although the 2-mode parametric retrieval indeed performs somewhat better for most parameters on the 10 mode synthetic measurements than vice versa, overall we believe that the performance of different retrievals on inconsistent synthetic measurements is sufficiently good compared to the level-2 requirements (in the revised version a requirement table has been added following the comment of reviewer 1). The only exception is the fine mode refractive index for which the poor performance of the 10 mode retrieval on the 2 mode synthetic measurement is not understood (as mentioned in the paper).

3. *For the PARASOL retrievals in section 5 it is stated that multi-mode retrivals with more than 4 modes perform well (while at the same time mode-5 seems to have the largest bias), whereas the conclusion from the synthetic retrievals was that multi-mode retrievals perform*

*well for n>5. Could there be a reason for this (although small) discrepancy. Overall the results are presented as if mode-5 is a kind of physical significant lower limit for multi-mode retrievals (if yes, why?), while the results seem to more indicate a general trend for decreasing RMS with higher mode numbers.*

Response:
You are right. Only for synthetic retrievals we see for most parameters a decrease in error for increasing number of modes till 5 modes and after that only a small decrease. For the real retrievals indeed this conclusion does not hold and we have removed statements in the paper that suggest this.

4. *In Section 3.1 there is a reference missing to the actual PARASOL data and its version used. Are there references available for the expected intensity and polarisation error for PARASOL?*

Response:
We added a statement that we use level-1 Collection 3 data, in Sect. 3.1,
" The PARASOL level-1 Collection 3 product data have been used in this study. "

Information about the measurement errors of POLDER-3 is hard to find. The results of Fougnie et al. (2007) seem to indicate (although not explicitly claimed) a radiometric error of about 2% for all bands except 443 nm where it would be about 5%. For the DoLP error no explicit reference exists. Knobelspiesse et al. (2012) assume 0.02 in their sensitivity study while Dubovik et al. (2011) use 0.005 but add the statement that the error is likely to be a factor 2-3 larger. Given that the exact magnitude of the POLDER-3 errors is of less importance for our study, we decided not to speculate on the magnitude of the errors.

The revised manuscript starts from next page.

[revised manuscript text omitted]
_{\text{eff}}$ ($\mu m$) | 0.070 | 0.094 | 0.130 | 0.163 | 0.220 | 0.282 | 0.882 | 1.2 | 1.759 | 3.0 |
| $v_{\text{eff}}$ | 0.130 | 0.130 | 0.130 | 0.130 | 0.130 | 0.130 | 0.284 | 1.0 | 1.718 | 1.718 |
| $f_{\text{sphere}}$ | | | 1.0 | | | | | | 0.5 | |
| $m_{\text{r}}$ (550 $nm$) | | | | | 1.45 | | | | | |
| $m_{\text{i}}$ (550 $nm$) | | | | | 0.02 | | | | | |
| $\tau$ | | | | 0.01, 0.15, 0.25, 0.5, 0.8, 1.0, 1.5, 3.0, 5.0 | | | | | | |
| $z$ (km) | | | | | 2.0 | | | | | |
| Wavelength bands (nm) | | | 390.0, 400.0, 410.0, 440.0, 450.0, 470.0, 491.5, 500.0, 550.0, 565.0, 600.0, 670.0, 750.0, 863.4, 1019.4 | | | | | | | |
| VZA (degree) | | | | 0, 10.0, 20.0, 30.0, 40.0, 50.0, 65.0 | | | | | | |
| SZA (degree) | | | 10.0, 15.0, 20.0, 25.0, 30.0, 35.0, 40.0, 45.0, 50.0, 55.0, 60.0, 65.0, 70.0, 75.0 | | | | | | | |
| Surface pressure (mbar) | | | | 700.0, 1013.0 | | | | | | |
| $x_{\text{bpdf}}^{\text{scale}}$ | | | | 1.0, 8.0 | | | | | | |
| $x_{\text{bdrf}}^{\text{geo1}}$ (LI kernel) | | | | 0, 0.1, 0.2 | | | | | | |
| $x_{\text{bdrf}}^{\text{geo2}}$ (ROSS kernel) | | | | 0, 0.5, 1.0, 1.5 | | | | | | |
| $x_{\text{bdrf}}^{iw}$, ($iw = 1, 2, \cdots, 11$) | | | 0.01, 0.015, 0.02, 0.03, 0.05, 0.07, 0.1 | | | | | | | |
| $x_{\text{bdrf}}^{iw}$, ($iw = 12$) | | | 0.01, 0.015, 0.025, 0.06, 0.075, 0.1, 0.125 | | | | | | | |
| $x_{\text{bdrf}}^{iw}$, ($iw = 13$) | | | 0.05, 0.07, 0.1, 0.13, 0.175, 0.25, 0.35 | | | | | | | |
| $x_{\text{bdrf}}^{iw}$, ($iw = 14, 15$) | | | 0.1, 0.15, 0.2, 0.25, 0.35, 0.4, 0.5 | | | | | | | |

**Table 7.** AERONET stations for validation of PARASOL retrievals

| | | | | |
|---|---|---|---|---|
| Alta_Floresta | Ames | BONDVILLE | Bac_Giang | Banizoumbou |
| Belsk | Cabauw | Chiang_Mai_Met_Sta | Fontainebleau | Fresno |
| Kanpur | Lille | Minsk | Mongu | Moscow_MSU_MO |
| Mukdahan | Trinidad_Head | Zvenigorod | XiangHe | Beijing |